# Rock and fault rheology explain differences between on fault and distributed seismicity

C. Collettini ®[1,2,3] ✉, M. R. Barchi ®[4], N. De Paola ®[5], F. Trippetta ®[1,3] & E. Tinti ®[1,2]

Analysis of seismicity can illuminate active fault zone structures but also deformation within large volumes of the seismogenic zone. For the $M_w$ 6.5 2016–2017 Central Italy seismic sequence, seismicity not only localizes along the major structures hosting the mainshocks (on-fault seismicity), but also occurs within volumes of Triassic Evaporites, TE, composed of alternated anhydrites and dolostones. These volumes of distributed microseismicity show a different frequency-magnitude distribution than on-fault seismicity. We interpret that, during the sequence, shear strain-rate increase, and fluid overpressure promoted widespread ductile deformation within TE that light-up with distributed microseismicity. This interpretation is supported by field and laboratory observations showing that TE background ductile deformation is complex and dominated by distributed failure and folding of the anhydrites associated with boudinage hydro-fracturing and faulting of dolostones. Our results indicate that ductile crustal deformation can cause distributed microseismicity, which obeys to different scaling laws than on-fault seismicity occurring on structures characterized by elasto-frictional stick-slip behaviour.

In the upper 10–15 km of the continental crust, background micro-earthquake activity defines the seismogenic regime[1] where faults are mainly characterized by elasto-frictional behaviour[2,3]. Here, strain localizes along faults whose structure generally consists of a fault core, where most of the slip is localized, surrounded by a damage zone formed by widespread fractures and subsidiary small displacement faults[4,5]. The total fault zone thickness, including core and damage zone, scales with cumulative fault displacement[6]. However, for fault displacements larger than 2–3 km, fault zone thickness tends to remain constant at several hundreds of meters[7]. In the last two decades, improved techniques in earthquake location[8] and detection[9] have been used to image the in-depth structure of active faults at a resolution consistent with field geological observations[10–12]. In particular, the geometry of active faults at depth has been mainly illuminated by aftershock distributions[10], which define a region of high seismic activity near the activated fault[10,12]. This zone of enhanced seismicity includes and sometimes extends beyond the fault zone structure and

has been defined as a zone of shear deformation[13]. In terms of earthquake mechanism, the zone of shear deformation is characterized by an elasto-frictional rheology promoting stick-slip behaviour[14]. During the interseismic phase, or the stick phase, the fault is locked, and frictional healing allows for fault restrengthening and for the accumulation of elastic energy in the fault loading medium or within the zone of shear deformation[3,15]. When the shear strength is overcome, the velocity weakening frictional behaviour of seismically active faults favours frictional instability associated with earthquake slip with stress drop[3,15]. Following the mainshock, aftershocks relax stress concentration, and they are usually located at the rupture perimeter or along fault structural heterogeneities[15].

In the last twenty years, well-located aftershock sequences have highlighted peculiarities of fault structures like for example the contemporaneous activation of orthogonal strike-slip faults during the M7.1 Ridgecrest 2019 seismic sequence[16,17], or the planar and listric geometry of normal faults activated during the M6.3 2009 L'Aquila

[1]Dipartimento di Scienze della Terra, Università di Roma La Sapienza, Rome, Italy. [2]Istituto Nazionale di Geofisica e Vulcanologia (INGV), Rome, Italy. [3]Consorzio Interuniversitario Nazionale per la Scienza e Tecnologia dei Materiali, Firenze, Italy. [4]Dipartimento di Fisica e Geologia Università degli Studi di Perugia, Perugia, Italy. [5]Department of Earth Sciences, Durham University, Durham, UK. ✉e-mail: cristiano.collettini@uniroma1.it

sequence[18]. For some creeping faults, background microseismicity has been used to highlight parallel fault strands along the San Andreas fault near Parkfield[19], and the geometry of extensional detachments cutting the entire upper crust[20]. However, during a seismic sequence, seismicity is not necessarily exclusive of the major structures activated within the seismogenic layer. In the San Jacinto fault zone, most of the low magnitude seismicity occurs in a zone that is several kilometres wide at seismogenic depth[21]. Ridgecrest 2019 aftershock distribution highlights a 5–10 km wide zone around the main ruptures[22]. In some fluid pressure stimulations, a broad network of distributed fractures has been activated with no evidence for alignment along a major fault[23] and in central Italy during the M6.5 2016–2017 sequence ~30% of diffuse seismicity has been detected[24]. To explain this type of distributed seismicity several mechanisms have been proposed. These include, but are not limited to, fault step-over or fault branching[12,25], deformation accommodated by many small faults[23,25], a wide damage zone[12,21,26], loading from an ongoing ductile deformation[22].

Overall, these studies emphasize that fault structure, style of deformation and rheology play a primary role in controlling the distribution of seismicity. However, to test such hypotheses would require access to constrained geological observations, and geophysical and mechanical data from a single, active region, which is rarely available. Here we integrate geological and geophysical data with laboratory experiments on the rocks composing the seismogenic layer

of the Apennines to explain the significant amount of distributed seismicity observed during the M6.5 2016–2017 seismic sequence. To this aim, we adopt the following terminology: with "on-fault" seismicity we refer to aftershocks located within the fault structure that is activated by the mainshock and this fault structure contains the fault core, damage zone and at least part of the fault loading medium; with "distributed seismicity" we refer to abundant aftershocks occurrence within volumes of the crust not including major faults hosting mainshocks. Our results show that distributed seismicity can be explained by the coexistence of brittle and ductile rheology within the Triassic Evaporites, TE, a thick sedimentary succession composed of the alternation of anhydrites and dolostones.

## Results

### Geology and structure of the seismogenic regime

The area struck by the 2016–2017 Central Italy seismic sequence was affected by a Late Miocene-Early Pliocene compressional phase, with about N-S trending east-verging anticlines and west-dipping thrust faults. This compressional phase was followed by Late Pliocene–Quaternary extension accommodated along NW-SE trending normal faults (Fig. 1). The seismic sequence started with the $M_w$ 6.0 Amatrice earthquake on 24 August 2016 and was followed by the $M_w$ 5.9 Visso and $M_w$ 6.5 Norcia earthquakes on 26 and 30 October, respectively (Fig. 1a). These three mainshocks nucleated on a set of

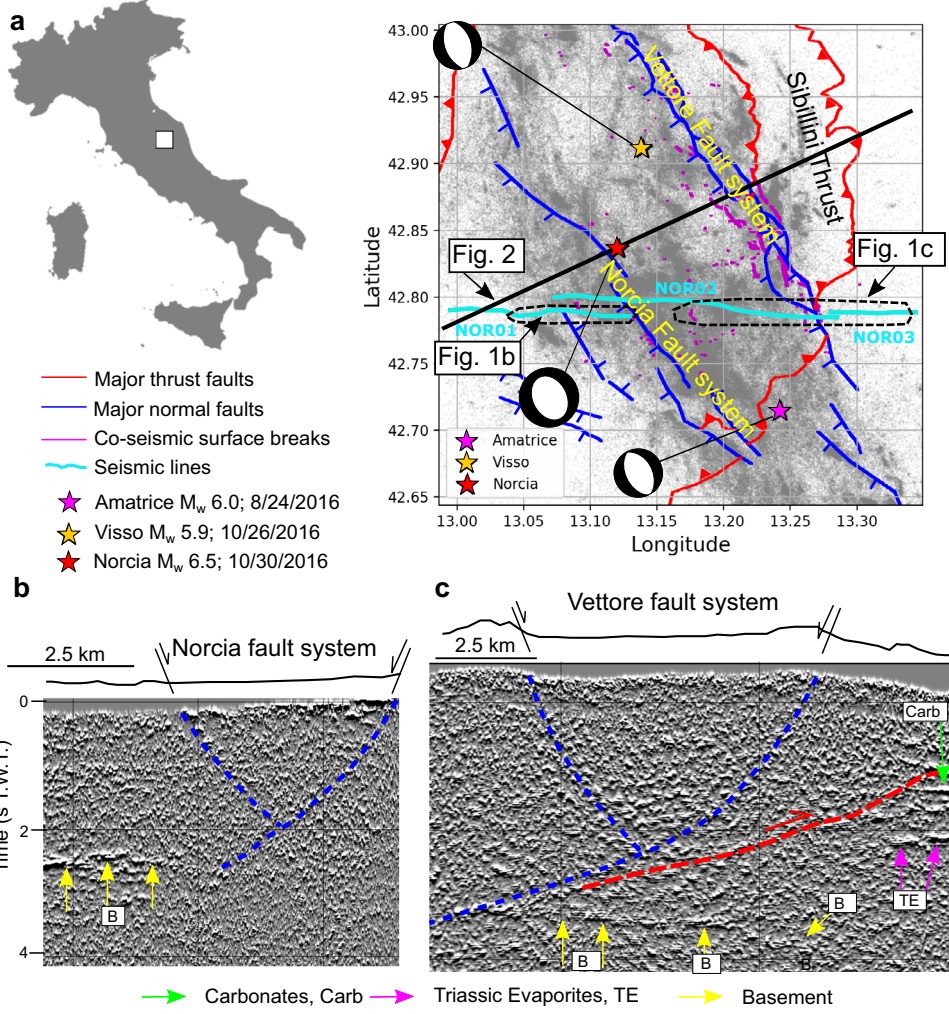

**Fig. 1 | Map of the sequence with surface and subsurface geology. a** Map view of the study area, grey dots represent located earthquakes[24]. Co-seismic surface breaks along the Vettore and Norcia fault systems[36], and moment tensor solutions[61].

**b, c** seismic images of the subsurface geology (seismic traces are reported in **a**). Blue dashed lines are the Norcia and Vettore fault systems at depth (details in Supplementary Note 1 and Supplementary Figs. 1, 2).

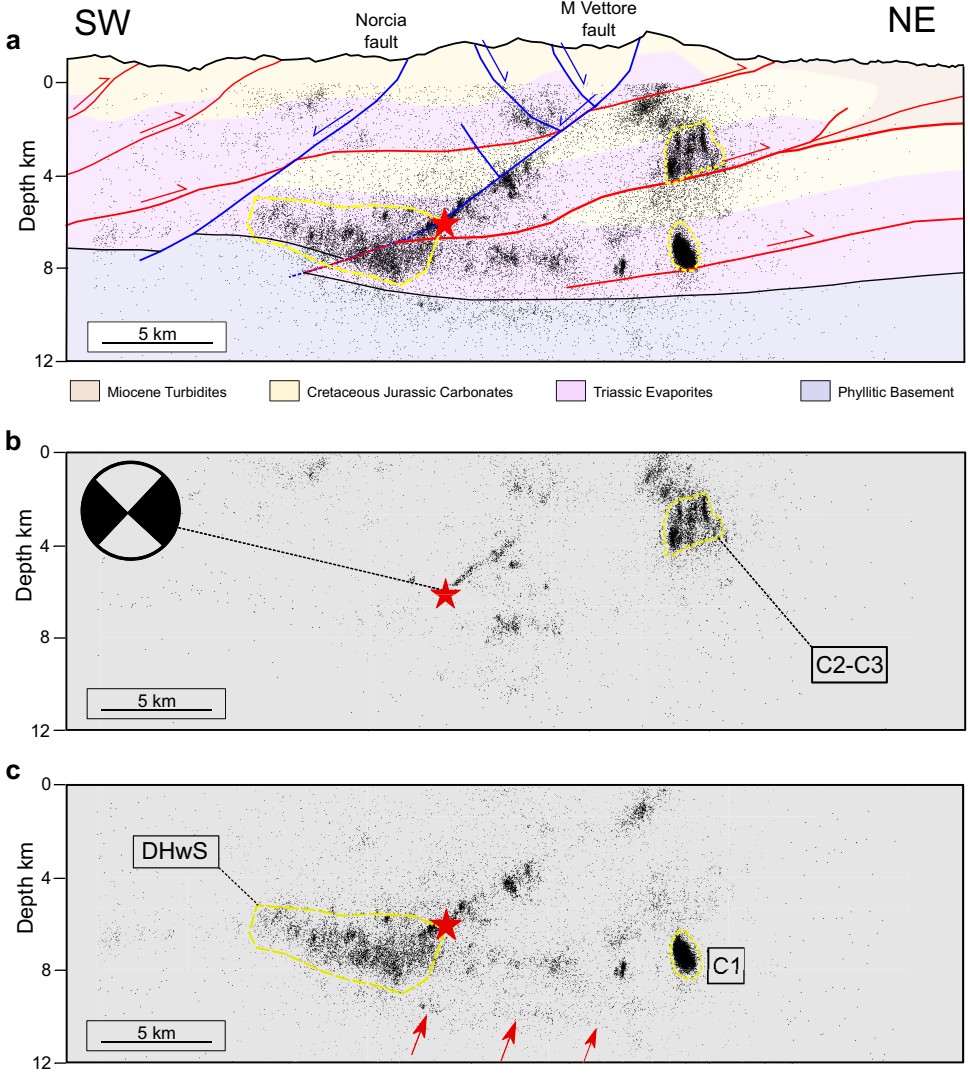

**Fig. 2 | Seismicity vs. geology. a** Cross section (trace in Fig. 1) integrating surface and subsurface geology (details in Supplementary Note 1 and Supplementary Figs. 1, 2) with the earthquake distribution (entire catalogue from 08/15/2016 to 08/15/2017). **b** seismicity from Amatrice (8/24/2016) to Visso (10/26/2016) mainshock and **c** after Norcia (10/30/2016) mainshock. Cross-sections are perpendicular to the strike (155°, from the moment tensor solution) of the Norcia mainshock (red star and moment tensor solution[61]). All events are within 1 km from the cross-section. Seismicity within sub-vertical clusters (C1-C3) and mainly located down-dip in the hangingwall of the Norcia mainshock, DHwS, is highlighted with dashed yellow lines. In **c**, red arrows at 9–12 km of depth mark an extensional shear zone presented in previous studies[27,29,30].

aligned SW-dipping normal faults[27–30]. The entire sequence activated an 80 km long, NW-SE trending normal fault system (Fig. 1a). The rocks composing the seismogenic layer in this portion of the Apennines are well constrained by seismic reflection profiles and deep borehole data[31,32]. In seismic profiles[33–35], the two major normal fault systems, Norcia and Vettore, are represented as steep alignments of disrupted reflectors that merge at the surface with mapped faults (Fig. 1). In the footwall of the M. Vettore fault, the structure that hosted the $M_w$ 6.5 earthquake and produced surface breaks[36], the integration of surface geology with seismic profiles has been used to reconstruct the compressional structures at depth ([35] and details in Supplementary Note 1 and Supplementary Figs. 1, 2). Figure 1c shows the geometry of one of the major thrusts of the area together with the reflectors of the carbonates and the TE, well-imaged in the thrust footwall. At greater depth, the top of the acoustic basement is located at 3.2 s Two Way Time, TWT (Fig. 1c), corresponding to 9 km of depth below sea level. The same reflector is imaged at 2.7 s TWT, corresponding to 7.5 km in the hangingwall of the Norcia fault (Fig. 1b). Close to the Norcia hypocentre the subsurface geology can be schematically represented by carbonates and TE at depths < 4–5 km and imbricated TE at depths between 5 and 9 km (Fig. 2a and Supplementary Note 1). The base of the imbricated TE coincides with the top of the acoustic, phyllosilicate-rich basement that is affected by compressional steps[35].

### Earthquake distribution

In this area, the presence of a dense seismic network and the application of improved earthquake detection and location techniques allowed the development of comprehensive earthquake catalogues[24,28–30]. The integration of subsurface geology with earthquake location well depicts the geometry of on-fault seismicity occurring on the SW-dipping Vettore fault, but it also highlights that a significant amount of seismicity is occurring within rock volumes of TE (Fig. 2a). The seismogenic volume affected by the 30 October Norcia $M_w$ 6.5 mainshock starts to be illuminated by microseismicity soon after the 24 August Amatrice mainshock. Here, the SW-dipping plane of the Norcia mainshock is highlighted by the microseismicity that occurred in the two months preceeding the Norcia event (Fig. 2b and ref. 29), from 6 km of depth (the hypocentral depth) to

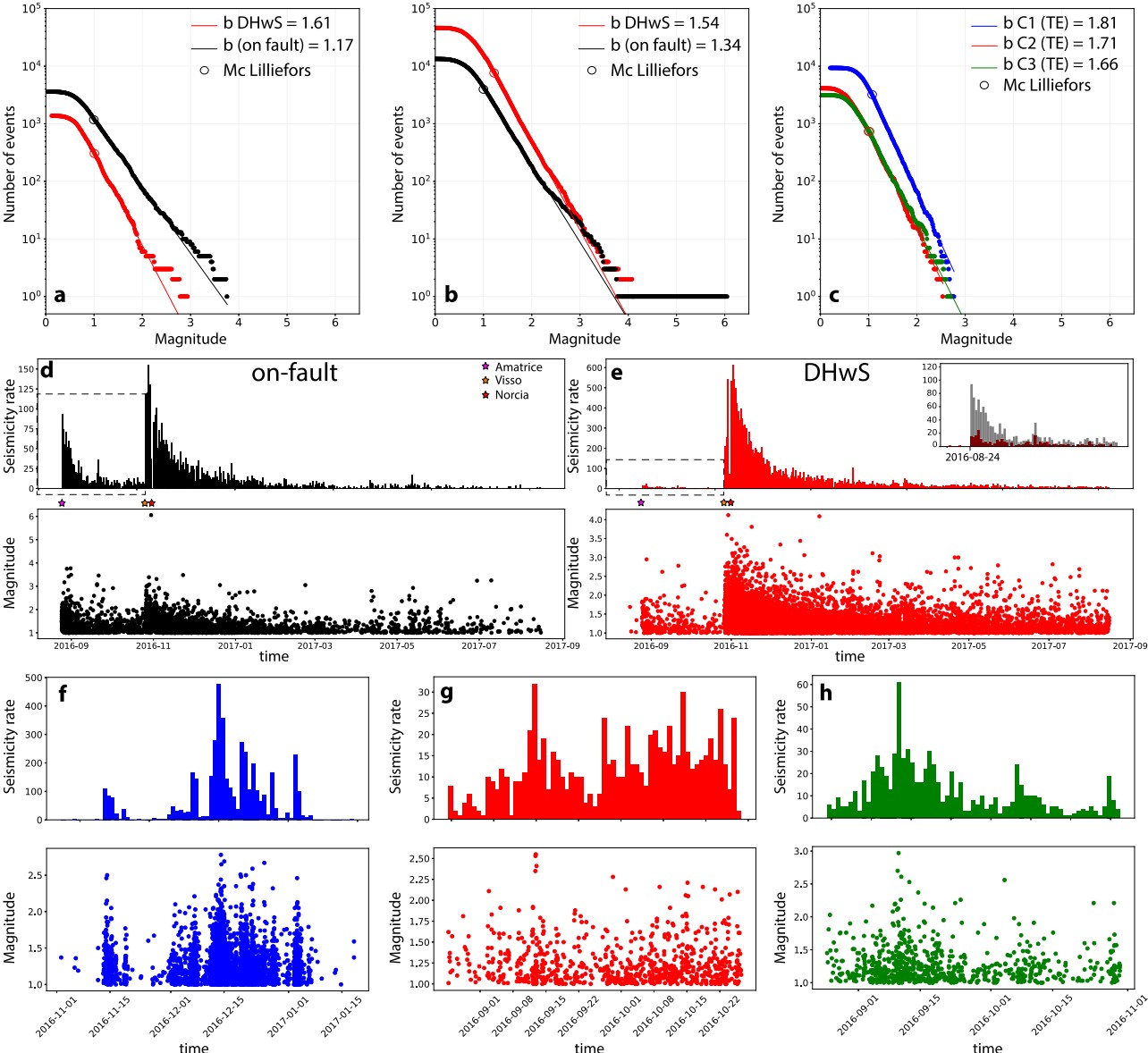

**Fig. 3 | On-fault vs. distributed seismicity. a** *b*-values for on-fault vs. down-dip hangingwall seismicity, DHwS, before the two major events of Visso-Norcia, and (**b**) for the entire sequence from 08/15/2016 to 08/15/2017. **c** *b*-values for the seismicity in clusters C1-C3. **d** Time evolution of the daily number of earthquakes and magnitudes for on-fault, DHwS (**e**) and clusters C1 (**f**), C2 (**g**) and C3 (**h**). The inset in (**e**) shows a detail of the seismicity rate before Visso-Norcia mainshocks for on-fault, in grey, and DHwS in red. In (**d**–**h**) we selected earthquakes with M > 1.0 consistently with Mc Lilliefors (see Method).

about 2 km. At depth > 6 km a few microearthquakes are located SW of the hypocentre whereas some small clusters are present NE from the hypocentre at about 8 km of depth. In this portion of the seismogenic volume, before the Norcia mainshock, significant earthquake activity is concentrated within kilometres long subvertical clusters at about 2-4 km of depth and located within TE (C2-C3 in Fig. 2a, b) and the carbonates above. Following the Norcia mainshock seismicity distribution still highlights the SW-dipping plane hosting the mainshock and merging at the surface with the Vettore fault (Fig. 2a, c). However, together with this on-fault seismicity, the seismogenic volume is also characterized by distributed seismicity. We observe a 6-8 km wide and up to 4 km thick zone of seismicity mainly located down-dip in the hangingwall of the mainshock rupture: we will subsequently refer to this seismicity as, down-dip hangingwall seismicity (DHwS in Fig. 2c). This seismicity extends both toward NNW and SSE along the strike of the activated Monte Vettore fault forming imbricated bands located within the TE that rest on top of the basement (Supplementary Note 2 and

Supplementary Fig. 3). We also observe the activation of other subvertical clusters of seismicity occurring in TE and located in the footwall of the Vettore fault, like the largest one located at about 8 km of depth (C1 in Fig. 2c). Following Norcia mainshock, seismicity is not present anymore in clusters C2 and C3 (Fig. 2c). Finally, in some areas of the seismogenic regime microseismicity alignment highlights a gently eastward dipping structure (red arrows in Fig. 2c) interpreted by previous authors as an extensional shear zone or a detachment[29,30]. This structure is more evident at depths of 9-12 km in the SE portion of the sequence (Supplementary Note 2 and Supplementary Fig. 3).

## Frequency-magnitude distribution
After reconstructing the geometry of the activated portions of the seismogenic layer and constraining the nature of the rocks involved in active deformation, we now analyse the frequency-magnitude distribution for on-fault vs. distributed seismicity. We search for any systematic variation of the *b*-value, which is the seismic parameter that

quantifies the proportion of small- to large-magnitude events[37–39]. For evaluating the *b*-value representative of on-fault seismicity we selected events within 0.5 km from the activated fault plane and at depths ranging from 6.1 to 2 km. This is consistent with the thickness and depth-range of the earthquake fault as imaged by aftershock distribution (Fig. 2) and is also in agreement with co-seismic fault slip, that only occurs up-dip from the nucleation point (Supplementary Note 3 and Supplementary Figs. 4, 5). Distributed seismicity was determined by selecting earthquakes occurring within TE in both the DHwS and within the clusters C1-C3 (Fig. 2 and Supplementary Note 3 and Supplementary Figs. 4–6). The *b* value is calculated using the revised maximum likelihood estimate (Methods and[40,41]). Our results show that *b*-values for on-fault events are different and systematically lower than those obtained for distributed seismicity. *b*-values of on-fault and distributed seismicity in DHwS are respectively 1.17 and 1.61 before Visso-Norcia mainshocks (Fig. 3a), and respectively 1.34 and 1.54 for the entire seismic sequence (Fig. 3b, and Supplementary Table 1). Widening the on-fault at 1 km or extending the DHwS of ± 1 km along strike yields essentially the same results. The *b* value for clusters hosted in TE is high and in the range 1.66-1.81 (Fig. 3c, and Supplementary Table 1).

Further differences between on-fault vs. distributed seismicity can be gained by the time evolution of the daily number of earthquakes and magnitudes (Fig. 3d–h). In clusters C1-C3 the largest events, with magnitude of about 2.5-3.0, are homogeneously distributed in time and the daily seismicity rate shows multiple increase and decrease through time (Fig. 3f–h). These trends are consistent with a swarm-like evolution[42]. For the on-fault seismicity the largest earthquakes occur soon after the mainshock and the evolution of the daily seismicity rate decreases with time following the mainshock aftershock Omori law[42] (Fig. 3d). After the Visso-Norcia mainshocks, the DHwS shows an evolution in time similar to the on-fault seismicity (Fig. 3e) that can be explained by the shear stress increase[37,38] affecting the DHwS area after the Visso-Norcia mainshocks[28]. Before the Visso-Norcia mainshocks, the DHwS is characterized by a nearly constant seismicity rate and evolution of magnitudes in time (Fig. 3e inset), together with a larger *b*-value (Fig. 3a), in agreement with what observed for the TE clusters.

Distributed seismicity is also present within the carbonates in the footwall of some major compressional structures like the two thrusts located at about 2 km and 4-5 km of depth, respectively, in the footwall of the Vettore normal fault, (Fig. 2a). However, this seismicity is not well-clustered in space and time as the one observed within TE (Supplementary Note 4 and Supplementary Fig. 7) and therefore not considered in our analysis.

## Rock and fault rheology

In this paragraph we are merging structural geology observations performed on outcrops of TE with rock deformation experiments on the same rocks to characterize differences in rheology between on-fault and bulk deformation of TE. In the study area, the TE formation consists of a thick, mechanically complex sedimentary succession composed of centimetric- to decametre-scale interbeds of Ca-sulphate rocks, gypsum predominantly at depth < 1 km and anhydrite at greater depths[43], and dolostones. Seismic profiles and boreholes show that the average thickness of the TE succession is ~2 km, but it can increase up to 4 km due to folding and thrusting (Fig. 2a and[35]). In the seismically active area of the Apennines TE do not crop out and have been drilled only in few deep boreholes[44], whereas to the west of the active area, in western Umbria and in Tuscany, outcrops of TE are well-exposed in a series of quarries[43].

To describe deformation observed in the TE in both outcrops and experiments, we use the following terminology. Ductile deformation refers to distributed deformation accommodated via folding (Fig. 4a, b) or distributed failure (Fig. 6d) without bulk stress drop (Fig. 6a blue and red curves). Brittle deformation refers to discrete and localized failure accommodated along fractures (Fig. 4f) and faults (Fig. 5), which display elasto-frictional behaviour and stress-drop (Fig. 6a black curves and c).

TE outcrops show a complex style of deformation, across a range of scales. At the hundreds of meters scale, TE show ductile deformation represented by folding of the gypsum/anhydrite and boudinage of the dolostones layers (Fig. 4a, b). Folding is highlighted by gneissic transposed fabric (Fig. 4b, c), which derives by the superposition of tectonic fabrics on the earlier compositional layering. Folding in the anhydrite layers produces fractures and domino-like structures in the dolostone layers (Fig. 4c, d). These rotated faults in dolostones detach into gypsum/anhydrite rocks (Fig. 4d), emphasizing the interplay between (brittle fracturing and faulting) and ductile (folding) in the rheological heterogeneous TE. Small displacement normal faults are also present at the boundary between gypsum/anhydrite rocks and dolostones (Fig. 4e). Intense subvertical hydrofracture systems (Fig. 4f) and small displacement normal faults are documented within the larger dolostone blocks. The intense hydrofracture systems point to brittle processes promoted by fluid-pressure fluctuations during TE deformation[43].

Large displacement (>100 m) normal faults (Fig. 5) are characterized by a fault core where most of the slip is localized along fault parallel principal slipping surfaces made of a fine-grained, dolomite-rich cataclasite[40]. The damage zone of major faults consists of foliated (fault-parallel foliation) gypsum/anhydrite rocks and heavily fractured dolostones (Fig. 5). These field observations emphasize the bimodal style of deformation for TE. Away from the major normal faults the deformation is pervasive and mainly controlled by the ductile behaviour of the anhydrites, brittle processes are limited within the dolostone layers or along small displacement normal faults. Along the major normal faults, the deformation is brittle, and the fault zone structure has the typical geometry and rock fabric of the faults of the elasto-frictional regime[2,3].

Further insights into the bimodal style of deformation of TE can be obtained from rock deformation experiments. Here experiments on dolostones and anhydrite intact rocks are used to characterize the rheological behaviour of TE away from the major fault zones, whereas friction experiments on granular fault rocks provide details for on-fault deformation. Dolomite brittle behaviour is documented in a series of triaxial tests at pressure and temperature conditions equivalent to those present at seismogenic depths in the Apennines[45]. The rheology of anhydrite samples, collected from deep boreholes in the TE of the Apennines, is shown in triaxial loading tests conducted at constant confining pressure, $P_c = 100$ MPa, and different levels of fluid pressure, $P_f = 60, 80, 90$ MPa (Fig. 6a and[46]). At low effective pressure, $P_e = P_c - P_f$, or for very high fluid pressure levels (black curves in Fig. 6a), after yielding and a phase of deformation at constant differential stress the sample undergoes brittle failure with a sudden stress-drop and the development of a localized fault and a thick, 1–2 mm, gouge layer. At higher effective pressure (blue and red curves in Fig. 6a), after yielding, the sample undergoes ductile failure at constant differential stress with no sudden stress drop and the development of a pervasive network of distributed shear bands (Fig. 6d). The ductile behaviour of the anhydrites also promotes very low-values of permeability, that are maintained low ($10^{-19}$–$10^{-21}$ m$^2$) even during the ductile deformation of the rock[46]. Rock deformation tests confirm the propensity of the anhydrites for a ductile behaviour, and this is strongly consistent with the bulk style of deformation of TE observed in the field (Fig. 4). High pore fluid pressure conditions, favoured by the low-permeabilities of the anhydrites, can also cause hydrofractures within dolostones (Fig. 4f) and localized brittle failure with stress drop within the anhydrites (Fig. 6a).

Large displacement normal faults within TE show fault rock assemblages of cataclasites and fault gouge, typical of the elasto-frictional regime (Fig. 5), where friction plays a key-role in fault

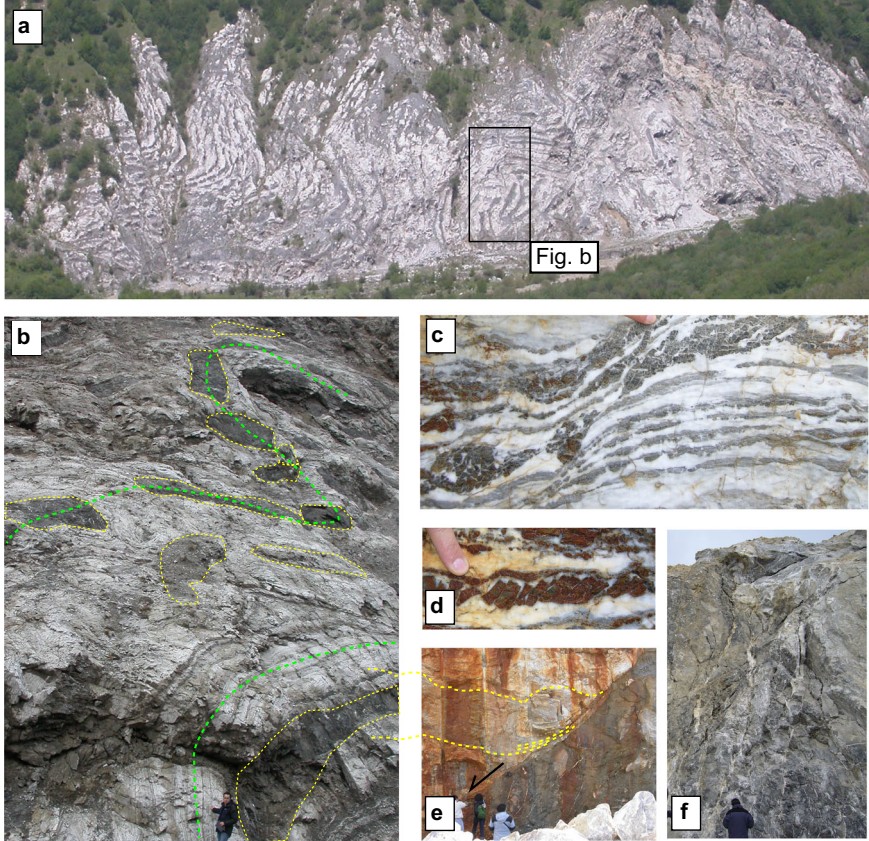

**Fig. 4 | Bimodal style of deformation of Triassic Evaporites, TE. a, b** Foliated and folded gypsum/anhydrite rocks (white) with boudinaged dolostone (grey lenses highlighted by dashed yellow lines). The dashed green line marks the geometry of the folds. **c** Gneissic transposed fabric affected by normal faulting and boudinage of the dolostone layers. **d** Domino-like structure with brittle faulting on dolostones and ductile deformation on gypsum/anhydrite. **e** Small displacement (meters), gently dipping normal fault at the boundary between gypsum/anhydrite rocks. In the hangingwall dashed yellow lines mark the foliation within gypsum/anhydrite rocks whereas fractured dolostones are present in the footwall. **f** Intense sub-vertical hydrofracturing within the dolostones.

rheology. Friction tests on anhydrite-dolomite fault gouges show a linear relationship between normal and shear stress, in agreement with a brittle failure envelope[47]. Anhydrite-dolomite fault gouges also show significant fault healing and velocity weakening behaviour (Fig. 6b). This type of frictional properties indicates that TE fault cores, like those observed in the field (Fig. 5), can gain elastic strain energy when locked during the interseismic cycle, and promote frictional instabilities when, during tectonic loading, frictional strength is overcome. Frictional instabilities are frequently observed on these fault gouges (Fig. 6c and[48]). The instabilities are facilitated by grain-size reduction and localization along dolomite-rich principal slipping surfaces (Fig. 6e), similar to those observed in the field (Fig. 5).

## Discussion

The integration of geological and seismological data shows that, during the 2016–2017 Central Italy seismic sequence, seismicity occurs both on-fault, i.e., on SW-dipping normal faults[27–30] and within rock volumes (Fig. 2).

At depth greater than 6 km, i.e., below the hypocentre of the Norcia mainshock, the seismicity is concentrated on 2–4 km thick, sub-horizontal (Fig. 2) and imbricated bands (Supplementary Note 2 and Supplementary Fig. 3). Sub-horizontal aftershock geometry and extensional focal mechanisms have been used by previous authors to propose that this seismicity represents an extensional detachment[27] that in some places is fragmented[30]. From the imbrication of the seismicity bands, previous studies suggested the reactivation of ancient thrust faults formed during the Late Miocene-Early Pliocene compressional phase[28]. Here we integrate seismological, mechanical,

surface and sub-surface geological data to propose an alternative interpretation: the identified thick zones of microseismicity do not highlight the reactivation of a major fault at depth (i.e., an extensional detachment or an inherited thrust), but they instead represent volumes of distributed microseismicity within the TE. Imbricated seismicity bands, that are up-to 4 km thick, are present at depths between 5–9 km (longitudinal sections 6 and 7 in Supplementary Fig. 3). The base of the imbricated bands coincides with the top of the basement that is affected by compressional steps, i.e., thrusts rooted into the basement, formed during the Late Miocene-Early Pliocene compressional tectonic phase[35]. In our interpretation, these seismicity bands of distributed microseismicity at 5–9 km of depth are due to ductile deformation within Triassic Evaporites resting on top of the basement (sections 6 and 7 in Supplementary Fig. 3). SE of the Norcia mainshock, these zones of distributed seismicity are confined at depth by continuous seismicity alignments indicative of an extensional detachment (Fig. 2c and cross sections 4–5 in Supplementary Fig. 3), in accord with the previous studies[27,29,30]. In other portions these zones are confined at depth by the top of the basement (Supplementary Note 2 and Supplementary Fig. 3), where frictionally stable, foliated, and phyllosilicate-rich horizons favour aseismic deformation[49].

Our integrated dataset explains the observed seismicity by a bimodal deformation regime, with on-fault seismicity due to localized deformation and elasto-frictional behaviour along the major normal faults of the area, and distributed seismicity due to pervasive and predominant ductile shearing within the TE (Fig. 7). On-fault deformation is well imaged near the Norcia mainshock where earthquake distribution well-depicts the geometry of the activated SW-dipping

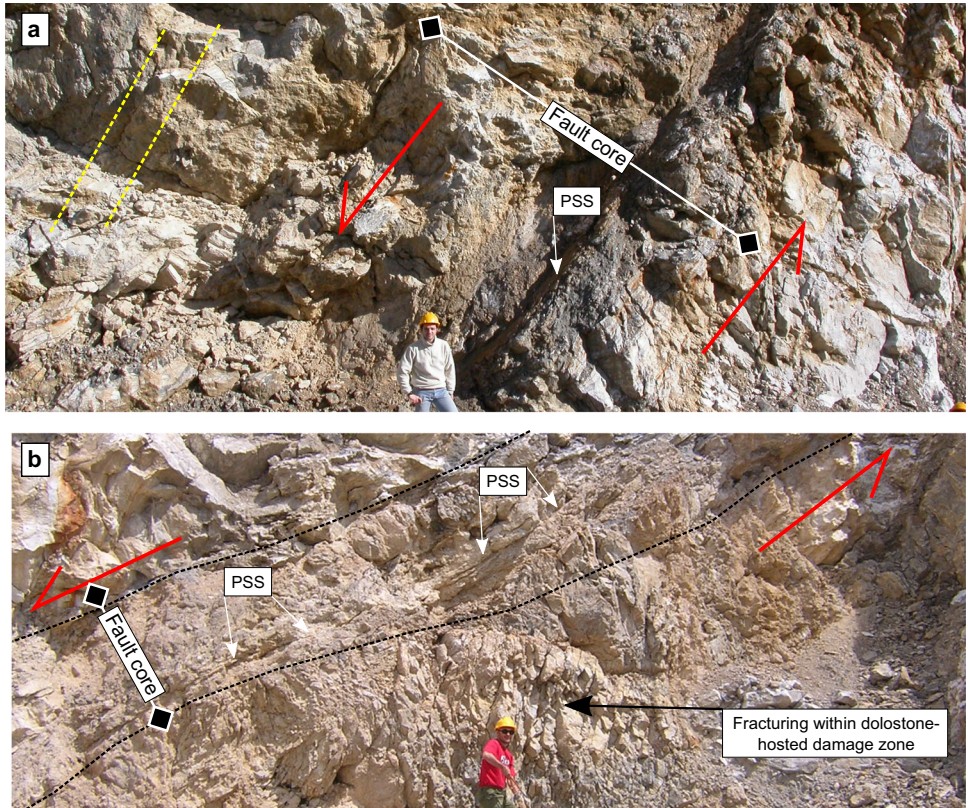

**Fig. 5 | Brittle faulting along major normal faults within Triassic Evaporites, TE.**
**a**, **b** Large displacement (hundreds of meters) normal faults with brittle deformation, characterized by grain-size reduction and localization along principal slipping surfaces, PSS. The dashed yellow line marks the fault parallel foliation within the gypsum-anhydrites rocks.

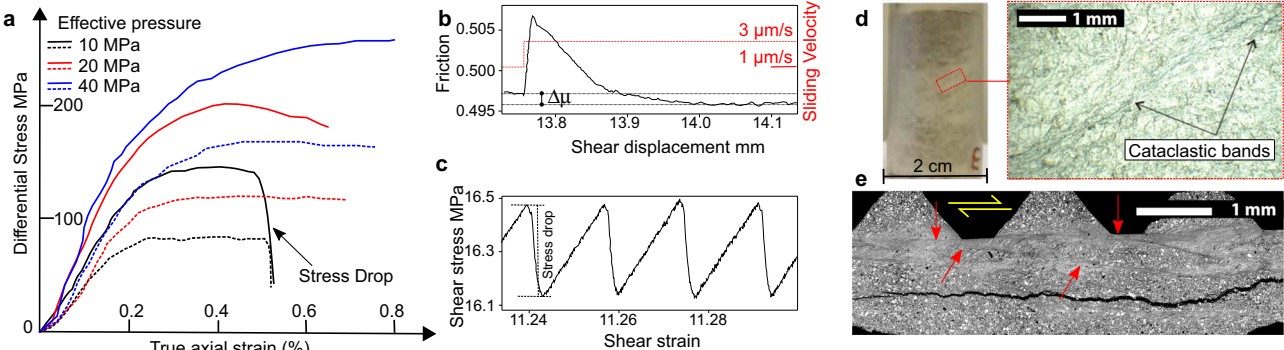

**Fig. 6 | Rock vs. fault rheology. a** Summary of triaxial tests on anhydrites cylindrical samples. Experiments were conducted at constant confining pressure, $P_c = 100$ MPa, and fluid pressure, $P_f$, of 60, 80 and 90 MPa, with a resulting effective pressure, Pe = $P_c$ -$P_f$, 40, 20 and 10 MPa. Dashed and solid lines for axial loading parallel and orthogonal to the foliation respectively. Anhydrite deformation is ductile and turns into brittle for high values of fluid pressure. **b** Frictional rheology of anhydrite-dolomite fault gouge: reduction in friction, $\Delta\mu$, following a velocity step (from 1 to 3 µm/s) resulting in a velocity weakening behaviour. **c** stick-slip cycles with shear stress drop of about 0.35 MPa. **d** Ductile deformation of anhydrite rocks during triaxial tests produces distributed cataclastic bands whereas stick-slip instabilities on anhydrite-dolomite fault gouges are favoured by slip localization (red arrows in **e**). Geological samples for experimental tests were collected in a responsible manner and in accordance with relevant permits and local laws.

normal fault[27–30], which can be followed with continuity from about 6 to 2 km of depth (Fig. 2). This on-fault seismicity occurs on large normal faults hosted within the carbonates and TE of the Apennines, showing fault structure and fault rocks typical of the elasto-frictional regime[43,50–52]. Along these structures deformation is localized (Fig. 5), and fault frictional properties are prone to promote earthquake nucleation via their stick-slip behaviour (Fig. 6c).

Distributed seismicity occurring within volumes of TE has been observed predominantly down-dip in the hangingwall of the mainshocks seismic rupture (DHwS in Fig. 2), and in kilometres long subvertical clusters at different crustal levels (C1-C3 in Fig. 2). Within C1-C3 the seismicity is concentrated in one or two months, the largest events (M ~ 2.5–3.0) are homogeneously distributed in time, and the daily seismicity rate shows multiple increase and decrease through time (Fig. 3f–h). We propose that the observed clustered seismicity is produced during the sequence by the destabilization of mechanically heterogeneous TE with compartmentalized fluid pressures. This seismic activity is favoured by the ductile aseismic behaviour of the

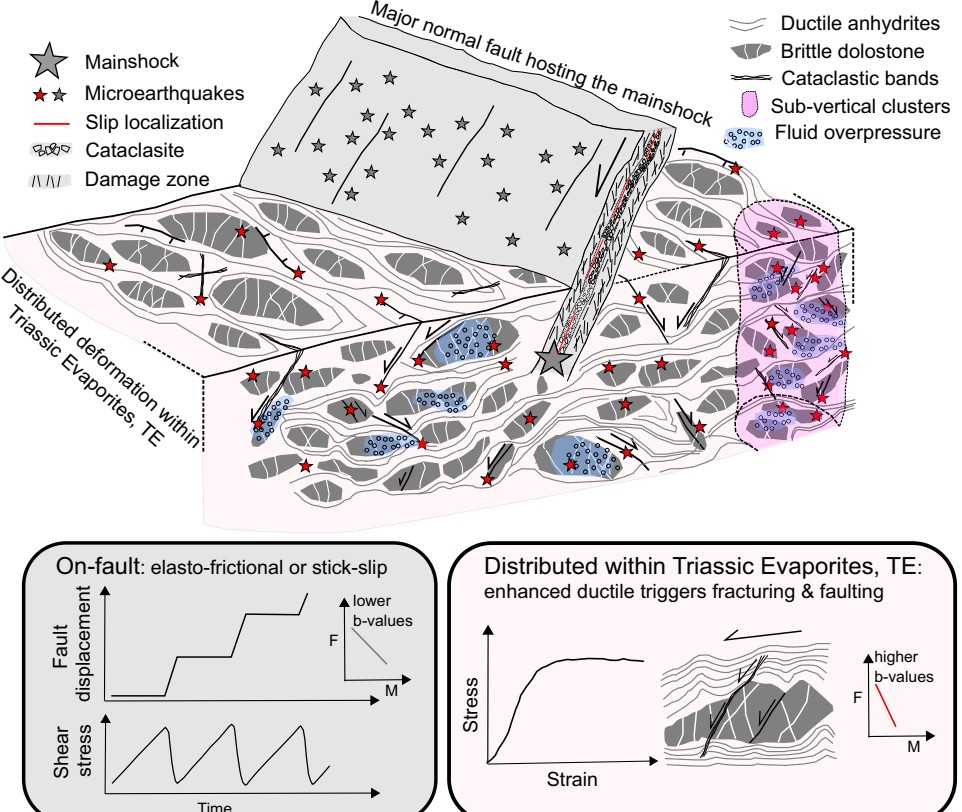

**Fig. 7 | The role of rheology for on-fault vs. distributed seismicity.** Cataclasis and slip localization along major normal faults produce fault rocks with frictional properties that promote earthquake nucleation via their stick-slip behaviour. Within the Triassic Evaporites, TE, during the seismic sequence enhanced ductile deformation of the anhydrites and fluid overpressure favour distributed fracturing and faulting of dolostones, brittle failure of anhydrites and reactivation of small displacement faults. On-fault and distributed seismicity are reported with grey and red stars respectively. Distributed deformation on large rock volumes of TE is associated to higher *b*-values of the earthquake frequency(F)-magnitude(M) distribution whereas on-fault seismicity is characterized by lower *b*-values.

anhydrites (Fig. 6a), with distributed ductile failure and folding, and associated boudinage of the dolostone rock bodies (Figs. 4 and 7). The very low permeabilities[46] of the anhydrites also facilitate fluid pressure development promoting a general embrittlement of the rock (Fig. 6a). This interpretation is supported by the intense hydrofracture system observed in the field within the dolostones (Fig. 4f), the high Vp/Vs anomalies observed during the sequence in correspondence of C2-C3[53], and the fluid overpressure at ~ 85% of the lithostatic load measured within the TE in two deep (4–5 km) boreholes[54] within the active area of the Apennines. In particular, the Pieve Santo Stefano borehole, shows nine thin levels (10–20 m) of compartmentalized fluid overpressures within dolostones that are sealed by anhydrites[44]. The DHwS is located down-dip and predominantly in the hangingwall of the Norcia mainshock, it appears after the mainshocks (Fig. 2), and it shows an evolution in time like on-fault seismicity (Fig. 3e). We propose that following the mainshock, shear strain-rate increase[3,55] and the development of fluid overpressure patches promoted brittle and ductile failure in these volumes of TE, which light-up with diffuse microseismicity (Fig. 7). In a similar way to C1-C3, this microseismicity is the result of the development of fracturing and small brittle faults in the dolostones, distributed ductile failure in the anhydrites and reactivation of small displacement, gently-dipping, minor normal faults.

Further rheological differences between on-fault and distributed deformation can be inferred by the analysis on the *b*-values. In general, *b*-value shows an inverse dependence on differential stress[37–39], it increases on increasingly rough faults[56], and during earthquake swarms high *b*-values are linked to fluid diffusion and reactivation of numerous small faults[25]. For some seismic sequences a near real-time characterization of the *b*-value has been used to discriminate between foreshocks

(decreasing *b*-values) and aftershocks (increasing *b*-values)[57]. However, the influence of structural complexities and expert judgment on the outcome of the analysis limit the use of *b*-value evolution for earthquake forecasting[58,59]. In this work, we show how *b*-value analysis can be affected by and used to highlight heterogeneous rock and fault rheology (Fig. 7). Our work shows that distributed microseismicity within TE is coupled with *b*-values that are significantly higher $1.54 < b < 1.81$ than those obtained for on-fault seismicity $1.17 < b < 1.34$ (Fig. 3). We suggest that the lower *b*-values for on-fault seismicity likely reflects the elasto-frictional deformation expected along the major structures of the crust hosting the mainshocks and characterized by stick-slip behaviour. We interpret the higher *b*-values, observed for distributed seismicity within the TE, as the result of both the ductile-brittle behaviour of the TE and the propensity of the anhydrites to trap crustal fluids and favour fluid overpressures. These strong heterogeneities in rock rheology and stress distribution favour the activation of a large number of distributed faults and fractures with limited size (Fig. 7).

Our results highlight the strongly heterogeneous nature of crustal deformation, emphasizing that a significant number of microearthquake activity during seismic sequences can occur away from the main activated structures and within large rock volumes. Ductile crustal deformation can cause distributed microseismicity, which obeys to different scaling laws than on fault seismicity. Lithological heterogeneities in the rock units composing the seismogenic layer[60] strongly influence seismicity distributions and seismicity rates. Our findings show that rheological behaviour of crustal rocks needs to be considered to explain the complexities of seismic sequences and advance our understanding of earthquake physics, including earthquake scaling laws.

## Methods

The frequency-magnitude distribution of earthquakes is usually modelled with an exponential function, called Gutenberg-Richter law, written as: ln $N(M) = a - bM$, where N(M) represents the number of events with magnitude larger than M, $a$ is the productivity and $b$ controls the relative rate of small and large earthquakes. Estimating the $b$-value appears trivial in theory but not in practice[41,58]. From the Gutenberg-Richter law, the probability density function of M is $f(M) = b\ln(10)\frac{10^{-bM}}{10^{-bM_{min}} - 10^{-bM_{max}}}$, where $M_{min}$ and $M_{max}$ are, respectively, the minimum and the maximum magnitude. For distributions that have $M_{max} - M_{min} \geq 3$ the probability density function can be simplified to $f(M) = b\ln(10)10^{-b(M-M_{min})}$. We derived the $b$-value with the Maximum Likelihood Estimation (MLE)[40,41] method according to the corrected formula[40] that accounts for the discrete nature of magnitude values:

$$b = \frac{1}{\ln(10)\left(<M> - \left[M_{min} - \triangle M/2\right]\right)} \quad (1)$$

where $<M>$ is the sampling average of the magnitudes, ΔM is the magnitudes binning or discretization (ΔM=0.01 for the adopted catalogue). The choice of the $M_{min}$ value in Eq. 1 is important to avoid severe bias in the estimation of the $b$-value. In this work we used the Lilliefors test that is a modification of the Kolmogorov– Smirnov (KS) test to assess whether the magnitude is exponentially distributed. Lilliefors is performed as a function of $M_{min}$ value for many initializations of the random noise (added to transform into a continuous random variable the binned magnitudes) from which we obtain a probability at each magnitude bin that expresses if the assumed null hypothesis is true (the exponential distribution). Through a recursive test the $M_c^{Lilliefors}$, or the Lilliefors-based magnitude of completeness, is defined as the lowest magnitude level above which the MFD can be considered exponential. For the probability distribution we used a significance level of 0.1.

Following the procedure described above, the $b$-value has been calculated starting from the high-resolution catalogue of ref. 24 and dividing the catalog into on-fault and distributed seismicity occurring within Triassic Evaporites (Fig. 2, Supplementary Note 3 and Supplementary Fig. 4). The Python Jupyter notebook we used to evaluate $b$-value can be found at Code availability.

For on-fault seismicity we selected all the events having a distance less than 0.5 km from the fault hosting the Norcia mainshock. The fault plane is defined with a strike of 155°, obtained from the mainshock moment tensor[61], and dip of 40°, inferred from aftershocks distribution, cf. for example Fig. 2b of the main text. We selected earthquakes at depth between 2 km and 6.1 km (hypocentral depth) where microseismicity distribution clearly shows the earthquake fault geometry (Supplementary Note 3 and Supplementary Fig. 4), and for a distance along strike direction less than 5 km (Supplementary Fig. 5). Beyond this distance the activated fault plane is not clearly imaged by aftershocks distribution.

For distributed seismicity within TE, we selected earthquakes occurring in:

a.  the thick zone of distributed seismicity mainly located down-dip in the hangingwall of the mainshock rupture, DHwS, and nucleating within the Triassic Evaporites (cf. Fig. 2 of the main text). This DHwS has been selected within a volume roughly approximated by a parallelepiped (details in Supplementary Fig. 5 and in the Jupyter notebook, see Code availability).

b.  kilometres long subvertical clusters of seismicity nucleating within Triassic Evaporites. One of these clusters (blue in Supplementary Fig. 6) is well-defined in space, the other two (red and green in Supplementary Fig. 6) represent a series of sub-vertical clusters that we merged to achieve a reasonable number of earthquakes for the $b$-value analysis. Since the inputs for this

space selection criterium are longitude, latitude and depth, a limited number of earthquakes belonging to these groups are not contained within the clusters.

Once defined the criteria to depict the geometry of on-fault vs. distributed seismicity, we also adopted a selection in time. For on-fault and DHwS we selected two time-periods the first one collects the seismic activity before the occurrence of the Mw 5.9 Visso event on 26 October and the Mw 6.5 Norcia event on 30 October. The second period extends from 15 August 2016 to 15 August 2017, corresponding to the entire catalogue published in ref. 24.The selected time windows for the clusters are related to the occurrence of the events composing the clusters: cluster C1 occurred mainly in December 2016 while C2 and C3 have been recorded during the two months preceding the Norcia main event (details in Supplementary Tale 1).

For any considered time-interval, from the earthquake catalogue, we excluded the short-term aftershock incompleteness (STAI), to avoid the bias due to this incompleteness. The Norcia mainshock has the strongest influence on STAI, and +2days of seismicity have to be removed whereas for the other mainshocks the influence is limited at +0.8, +0.6 and +0.4 days for Amatrice, Visso and Campotosto events respectively[59].

The number of events in each subset is large enough (>1500) to have stable $b$-values. The inferred $b$-values for all the considered subsets and time interval are reported in Supplementary Table 1 together with the 95% confidence intervals and $M_c^{Lilliefors}$ values. Widening the on-fault at 1 km or extending the DHwS of ± 1 km along strike yields essentially the same results.

## Data availability

The seismicity catalogue used in this work is published in Tan et al., 2021 and access can be obtained at the Zenodo dataset repository (https://doi.org/10.5281/zenodo.4662870). The triaxial deformation tests reported in Fig. 6a are from De Paola et al., 2009 and access can be obtained at Zenodo dataset repository (https://doi.org/10.5281/zenodo.6794379). Friction experiments presented in Fig. 6b, c belong to Scuderi et al., 2020, and access can be obtained at the Zenodo dataset repository (https://doi.org/10.5281/zenodo.3898725, experiment B396 and B518).

## Code availability

The Python Jupyter notebook for the analysis on the $b$-values can be downloaded at https://github.com/tintielisa/b_value_seismicity[62].

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

## Acknowledgements
We thank P. Ampuero, L. Chiaraluce, C. Marone, G. Pozzi and M. Scuderi for fruitful discussions. We thank Eni for providing us borehole samples of anhydrites and Lago d'Iseo quarry management in Roccastrada, Siena, for its constant availability.

## Author contributions
C.C. conceptualized the idea together with NDP. E.T. performed the analysis on *b*-values with conceptual inputs from C.C. C.C., M.R.B., N.D.P., F.T., and E.T. contributed to the data analysis, discussion, and interpretation of results. C.C. wrote the manuscript with the help of NDP and ET.

## Competing interests
The authors declare no competing interests.
