## [Peer review file · Nature Communications]

REVIEWER COMMENTS

Reviewer #1 (Remarks to the Author):

Review Collettini

The paper On fault vs. off-fault seismicity: the role of rock vs. fault rheology

by Collettini et al., brings a new interpretation of the diffuse seismicity in the Apenines, which does not occur along the main faults and shows higher b-values. They associate its occurrence to the heterogeneous structure of the rock medium, which is by part ductile and by part brittle.

The explanation of the diffuse seismicity by the heterogeneous geology fabric, which is supported by drillings, outcrops and rheology is novel and deserves publication.

The paper is interesting and reasonably well written. However, the first part on Geology and seismicity observation would benefit from some rewording, which also applies to the Discussion as it is not easy to read and understand. I also believe that adding the viewpoint of seismicity evolution in time of focal mechanisms would make the argumentation more robust and transparent. A moderate revision is, to my opinion required, to make the paper acceptable for publication.

GENERAL POINTS

1) The diffuse seismicity in dolostones and anhydrites could show variety of focal mechanisms than the on-fault earthquakes. The reason is the strongly heterogeneous structure of brittle dolostones encapsuled in ductile anhydrites and also fluid patches. So I believe tha analysis of focal mechanisms would definitely provide verification of the proposed model. However, It would require to extend the study, which is probably limited by length.

2) It is hard to uderstand the space relations between seismic clusters, faults and geological units in Fig. 2. I also think that the Supplementary Fig. 1 is unnecessarily detailed, compared to its sake - to show the occurrence of on- and off-fault seismicity.

3) While the seismicity analysis is convincing (but only after repeated reading), I got a bit lost in the geological part distinguishing anhydrites and dolostones and their rheological behaviour. On one hand there is a large number of photos showing the outcrops that iillustrate the rock rheology (I think less could suffice, but it I am mainly a seismologist...).On the other hand, I feel that the data, photos, and references provided (too many references which are rather shallow, without any explanations) could

allow for different interpretations of the off fault seismicity. Possibly fewer references could be given and focus to some of the references points in the text - this applies to the entire paper.

4) The authors distinguish between on-fault and off-fault seismicity by different b-value. I like this measure for seismicity classification; the b-value estimation looks well based. And this brings me to the fact that similar difference in b-values is observed for mainshock-aftershock sequences and swarms. And on top of that, mainshock and aftershocks usually occur on fault planes, but swarms can also be diffuse seismicity. I think the study would gain from incorporating this point of view. By checking other features typical for swarms to see if the off-fault seismicity is indeed missing mainshocks. This would require at least showing the magnitude-time plots. I also remind that the earthquake swarm occurrence is, despite to the fluid pressure propagation, also associated to the aseismic slip (creep) in a ductile environment, with brittle asperities distributed. And the Appenines diffuse seismicity might be this case.

PARTICULAR COMMENTS

75: Reference fo Miller et al (2004) appears not suitable here when speaking about seismic sections and geology

90: Even after repeated reading of this sentence and Fig. 2 I did not identify the 6-8 km wide and 4 km thick zone of diffuse seismicity.

104: The language is extremely brief and enigmatic - for example I find not acceptable to say 'but see also 33'. At least the basic message of the reference should be mentioned.

106 - 110: I did not get if a) and b) cases are somehow related to Fig 3a and 3b; besides the notation DHWS is not mentioned in Fig.3 caption

119: Referring to stress increase 39, 40 is again too brief - needs to be explained what stress increase and why. Perhaps indicating the Visso Norcia mainshocks in Fig. 2 in relation to the DHWS events could help.

137: What do you mean by 'elasto-frictional stress drop'? By comparing with the next line I find that probably 'non-zero stress drop would sound clearer'

188: What mainshock nucleation do you have in mind?

266: Why is the red off-fault seismicity limited from top exactly by 6.1 km? From the location plot it appears that also the events up to the 5km depth belong to it.

280: It would be very useful to see the time history of seismicity - magnitude vs. time - ideally for the on-fault and off-fault seismicity. This would help to see the time relation of the main earthquakes and also how the off-fault seismicity is related to the on-fault mainshock.

MINOR

79: 'acoustic basement' is not clear; this is the first occurrence. Change to 'phyllitic basement'

110: Supplementary Fig. 2-4 → Supplementary Figs. 2-4

144: (Fig 4c and c) sounds a bit unclear

FIGURES:

Fig. 1: ^{[[[]]} (b) and (b) should read as (b) and (c)

- map scale is missing in (b) right

Fig. 2: ^{[[[]]} it gets clear only multiple reading that 'from Amatrice (8/24/2016) to Norcia (10/30/2016)' is related to the time and not or space interval. It would be helpful to add the time interval to the plot titles. ^{[[[]]} is it right that Carbonates are beige colored and TE are rose? This should be clearly stated.

- to help reader get better oriented, different clusters could be color-coded, e.g. the DHWS events etc.

Fig. 4: ^{[[[]]} I think that the frame in (a) shows different rock than is zoomed in (b)

- figure f) is not referred to in the text

Fig. 6: (a) verb missing in the sentence Dashed and solid lines... (b) scale is missing in the bottom panel and the plot is too tiny, hardly readable

Supplementary figures

Figure 1: This graphics is too complicated to be understood without a very detailed and repeated reading of the text. Possibly using less sections, color coding of the hypocenters and more structured caption could help the reader to get the point. One question: Is the white=dash basement the same as phyllitic of acoustic basement mentioned before (also in Fig. 1). This should be unified; I found also other inconsistent namings.

Reviewer #2 (Remarks to the Author):

This paper explores how on- and off-fault seismicity might be controlled by the rheological behavior of the host rock in comparison to the frictional behavior along the fault. Using the Norica earthquake sequence, the authors map seismicity on and off fault, detail structures in the same rock types in the field, and run a set of strength and frictional experiments to compare on and off fault behavior. The authors conclude that diffuse off-fault seismicity is driven by ductile deformation off-fault that is triggered by on-fault seismicity and possible fluid pressure changes. Although it seems pretty likely that both ductile and brittle deformation is taking place here, as evidenced by the abundant examples in outcrop, the tie to the seismicity is less convincing. I think there are a lot of interesting things to unpack here, but some clarity and perhaps some caveats should be added to the paper. Below are the main issues that need to be addressed:

1) It is not convincing from Figure 2 that the diffuse seismicity is mostly limited to the Triassic evaporites (TE). The off-fault seismicity in the footwall containing the clusters studied in more detail (C1-C3) looks like it might be on vertical structure that goes through both TE and the carbonates. I would suggest looking at a time lapse of seismicity to determine whether this is actually an aseismic event that actually starts in the TE and then migrates into the carbonates. Furthermore, the authors point out that there is a lot of seismicity in the TE where the fault soles into a detachment, but is that not because there is a detachment there? Or are they arguing that a detachment is not required here? This is presented as an alternative hypothesis to a reactivated thrust fault, but is it not likely that an old thrust might be located within the anhydrite layer?

2) The authors show that the B values are lower on fault than off fault, which they attribute to off fault seismicity being driven by ductile failure. However b-value changes are terribly non-unique and an increase in b-values has been attributed to fluid pressure, fault size distribution, stress, and more. Could aseismic ductile deformation create a higher b-value? Sure, but we cannot prove that is what's happening here.

3) If there is a detachment fault that the more steeply dipping fault soles into, why is the detachment not included in the on-fault seismicity (Supplementary Figure 2)? This including some amount of that seismicity as being on-fault might change the b-values.

Reviewer #3 (Remarks to the Author):

See the pdf file attached.

General comment:

C. Collettini et al. propose to combine geological and seismological data to investigate the possible differences in seismicity distribution and associated laws (e.g., Gutenberg-Richter) following the Norcia-Amatrice earthquake sequences that occurred in 2016-2017. This original approach allows the authors to have a detailed picture of the fault structure thanks to the high-precision earthquake relocation catalog available in this area, and the link with the local geology which presents both brittle (carbonates) and ductile (evaporites, anhydrites) rheologies. They find that localized “on-fault” seismicity is mainly related to the main rupture of the fault plane in carbonates, responsible for the earthquakes. On the other hand, diffuse “off-fault” seismicity exhibits “higher” b-values than on-fault seismicity, and would be related to the rheological ductile behavior of evaporites.

While I found this study very interesting, associated with very nice data, I express below some major concerns that the authors should consider. They are related to different aspects of the study such as the definition of on-fault and off-fault seismicity or the lack of detailed descriptions of the methods (in particular because a large part of the work used in this paper has been published in previous studies). There are several typos in the text and figures and I suggest few improvements to add some clarity between figures.

Major comments:*Methods and definition of on-fault vs off-fault seismicity:*

I understand that the authors have to define some rules to select the seismicity. But I have some difficulties with their definition of on-fault and off-fault seismicity. Usually off-fault seismicity is the seismicity situated away from the fault plane (ie. perpendicularly to the fault strike and dip). And some papers have studied this seismicity distribution during interseismic and early postseismic periods, sometimes in direct link with geological observations (e.g., Powers and Jordan 2010; Hauksson, 2010; Perrin et al., 2021; Rodriguez Padilla et al., 2022). Here the off-fault seismicity considered by the authors is in the continuity of the fault plane along dip, which to me includes both parts of on-fault deformation rooted in a decollement layer and some off-fault deformation. The seismic profiles show that normal fault and thrust are merging at depth where the diffuse seismicity is observed. It seems hard to me to think that these fault planes are considered here as a “off-fault” zone selected by the authors (supp. Fig. 2). At least, the authors are not decorrelating both signatures in their selection.

In the same way, it seems that the width of the on-fault box is about 1km (or 2*500m each side from the fault plane, Supp. Fig 2), which is a significant volume where on-fault and off-fault processes are already occurring and could be discussed. For example, Perrin et al. (2021) have shown that diffuse off-fault deformation could occur up to 4-5 km away from the fault plane in case of immature fault cases (and the normal faults in the Apennines are likely immature).

I think it is important that the authors better describe in the text what they call on-fault vs off-fault zones to avoid any confusion with existing studies that are using the same words to describe different things (that also varies from tectonics to seismology...!). Maybe an alternate wording of what the authors measured would be “deformation in localized vs distributed fault sections”.

New contributions from this paper vs published studies and data availability statement:

I realize that the paper is a kind of synthesis where authors gather different pieces of a puzzle (seismology, geology, rheology) to get the whole picture. Thus, often the authors are referring here and there to published works (“see details in ...”). The problem with such type of references is that the reader as very little information on the data based on the single read of this paper. The point is not to do again the work from previous studies, but to give a minimum of information thus the paper strengths and weaknesses can be understandable quickly. In short, the paper should stand on its own. It would be useful for example to have information about error locations (absolute or relative locations), have some details on the formulas used: the method section is too vague. Please write the formulas and parameters you used rather than saying “we took one formula from here” (see also line by line comments). A dedicated detailed supplement could be useful.

In the same way, the data availability statement is not conformed to current publication policies: the supplementary material of this paper is actually not providing data but figures. The seismicity catalog used should be placed in an open archive or the author should cite the web address and associated references (zenodo link in Tan et al, 2021 paper). Same for the rheological tests, etc.

Interpretations

The idea that the rheology is impacting the seismicity location and frequency is really interesting. But I would go further in the comparison between off-fault and on-fault deformation in different rheological layers (see figure below).

Considering a single lithology affected by a fault (i.e., carbonates), a larger off-fault b-value is not incompatible with a homogeneous/low-fractured medium away from the fault plane compared to on-fault seismicity where you expect larger events on it.

Thus, I believe that a comparison between the different zones roughly circled in the figure below might strengthen the analysis.

Line-by-line comments and corrections:

l. 26: stick sip -> stick slip

l. 47: kilometres -> kilometers

l. 46-47: Well, the seismicity is off the main San Jacinto fault but orthogonal directions might be aligned with secondary faults and thus the seismicity be considered as “on-secondary-fault”. The mix between different scales of observations might be confusing for the reader, especially in the definition of diffuse off-fault deformation. In Ross et al., (2017), orthogonal faults are seen at the surface in the SW of San Jacinto but not in the NW, but it doesn’t mean that faults don’t exist at depth in the NW. In map view, the aligned seismicity patterns are not so diffuse taken individually as secondary faults. This case with orthogonal directions is pretty specific (like Supertition Hills/Elsinore earthquakes, Tottori, etc). Maybe this section needs to better explain more generally the different type of off-fault seismicity with more references (see major comment on the definition of on-fault and off-fault seismicity, damage zones, etc).

Fig1:

- wrong labels for 1c (b labeled twice)

- Do you have the possibility to have a better resolved fault map? or at least add symbols to show fault dip and slip motion.

- ruptures in light blue are barely visible on figure 1a

- Fig. 1b and 1c:

-> Looking at Barchi et al. (2021) and Ercoli et al. (2020), it seems that the two seismic profiles are actually continuous. Why did you cut them in two pieces? It would be better to have them in a single figure to easily compare with Figure 2.

-> Also, there are other east-dipping faults that you didn’t report and which could be associated with the cluster situated in the TE between 1-3 km depth between the Mt Vettore fault and the Norcia fault. This cluster is visible in the early times after the earthquake (figure 2b). That would be interesting to also analyze this “on-fault” secondary seismicity in TE to compare with other GR distribution (see also my major comment).

-> please specify in the caption that TE = pink dashed line and phB = yellow arrows. Why are you not using arrows everywhere (which would allow the reader to see the different reflectors)?

-> Barchi et al. (2021) and Ercoli et al. (2020) describe the TE reflector as a thrust. Why do you report it here as the top of evaporites? They also show in the eastern part of fig.2c that the reflectors above and below the thrust are actually carbonates so it is contradictory with the location of evaporites you are highlighting. How do you reconcile this with the geological interpretation in Fig.2? How confident are you in the stratification on which you depend a lot in your following analysis, especially for cluster C2 and C3?

Fig1 vs Fig 2: Please stay Consistent between your figures: the color of normal and reverse faults are inverted between Fig. 1 and 2. Please stick with one color code (normal= red ; reverse = blue).

- Fig. 2:

- > c label in the figure is surrounded by black arrows
- > in the figure caption, two 'b' labels are described and no 'c'.
- > specify that the red star is the Norcia mainshock
- > the use of i), ii) is confusing in a figure caption. Put this detail in the main text.
- > the seismicity seems cut above a depth of 0-0.2 km, which might depend on the method/constraints used for the relocations. But the topography is quite significative. Do you really expect no seismicity at shallow depth? or do you think you would need to refine the relocations/catalog? or consider a shift of the seismicity upward closer to the actual topography? Please discuss.

I. 88: off-faults seismicity -> off-fault seismicity ?

I. 88-89: figures are only showing cross sections (vertical axis = depth) so using both terms "wide" and "thick" is confusing. Please add a xlabel in your plots (for example "Across-strike distance") and use the corresponding term in your text for clarity. Also from your description, the diffuse seismicity seems to be rather 2 km thick, so please use depth range to spot exactly what you are describing.

I. 102-103: b-value is also sensitive to other parameters such as depth (but it is debated; e.g., El-Isa and Eaton, 2014) or fault segmentation in complex earthquake sequences like it is the case in your study area. In a preprint Herrmann et al are studying in detail this aspect (https://assets.researchsquare.com/files/rs-1210699/v1_covered.pdf?c=1641579587). I know it is not published (yet), but it might be good to anticipate and discuss this a bit more...

I. 105: 0.5 km: at which distance do you consider that your are not "on-fault" anymore? see also my major comments.

I. 107-109: I don't understand this criterion. Why considering as off-fault seismicity events situated down-dip in the direct continuity of the fault/aftershock sequence? This is a huge hypothesis, and the data (seismic profiles and geology) are not agreeing with that. See also my major comment.

I. 111-116: Don't you think that considering both zones as on-fault in carbonates and in evaporites might favor even more your interpretation on the impact of the rheology? Would you obtain the same results by selecting the seismicity in two wide boxes with the same size (i.e., same across strike width)?

I. 123-153: I like this part.

Fig. 4: the dashed red line is hard to distinguish.

Fig.6: label c hardly distinguishable among other strange characters around (problem in the export/conversion of the figures?)

I. 212: Maybe I missed something, but at this point it seems that you completely forgot to interpret the clusters C1, C2 and C3 situated in TE. They are not diffuse at all but well localized.

More discussions are definitely needed on this point. Is it the case in I. 220-222? If yes it has to be explicitly said and discussed.

I. 227: stick slip > stick slip

Fig. 7:

- nice sketch! However, it reflects part of the “reality”. From your Supp. Fig. 2, the seismicity in the TE ductile zone is not only situated in the hanging wall but also in the footwall. Then you measure a behavior in a distributed zone that encompasses both distributed on-fault and off-fault deformations.

- the left sketch shows clear small shear fault planes in the anhydrites but not in the dolostones (contrary to the right part). This might be confusing so please add some in the dolostones. Also, you don't explain the difference in the caption between grey (on-fault seismicity) and red (off-fault seismicity) stars.

- A suggestion: in the right parts of your figure, put in grey and red the on-fault and off-fault GR distribution to clearly present the link with the left parts of the figure (stars). Choose a different color for the mainshock otherwise we think that there is a link only with off-fault seismicity and not on-fault seismicity.

I.269: repetition of “defined”

I. 286: which is? what do you mean by excluded? Herrmann and Marzocchi (2021) show that aftershocks within 4 days after the mainshock present smaller b-values compared to the aftershocks considered in the following month (or more). What time period did you consider and/or remove?

I.287: Extimation => Estimation?

I. 288: which is? see also major comments

Supplementary Figure 1: Caption includes observations labeled twice a) and b) to describe two different things that are not labeled on the figure. This is very confusing. Please change this way to describe your figures.

References

- Barchi, M. R., Carboni, F., Michele, M., Ercoli, M., Giorgetti, C., Porreca, M., ... & Chiaraluce, L. (2021). The influence of subsurface geology on the distribution of earthquakes during the 2016–2017 Central Italy seismic sequence. *Tectonophysics*, 807, 228797.
- El-Isa, Z. H., & Eaton, D. W. (2014). Spatiotemporal variations in the b-value of earthquake magnitude–frequency distributions: Classification and causes. *Tectonophysics*, 615, 1-11.
- Ercoli, M., Forte, E., Porreca, M., Carbonell, R., Pauselli, C., Minelli, G., & Barchi, M. R. (2020). Using seismic attributes in seismotectonic research: an application to the Norcia $M_w = 6.5$ earthquake (30 October 2016) in central Italy. *Solid Earth*, 11(2), 329-348.
- Hauksson, E. (2010). Spatial separation of large earthquakes, aftershocks, and background seismicity: Analysis of interseismic and coseismic seismicity patterns in southern California. In *Seismogenesis and Earthquake Forecasting: The Frank Evison Volume II* (pp. 125-143). Springer, Basel.
- Herrmann, M., & Marzocchi, W. (2021). Inconsistencies and lurking pitfalls in the magnitude–frequency distribution of high-resolution earthquake catalogs. *Seismological Research Letters*, 92(2A), 909-922.
- Herrmann, M., Piegari, E., & Marzocchi, W. (2022). b-value of what? Complex behavior of the magnitude distribution during and within the 2016–2017 central Italy sequence, submitted pre print
- Perrin, C., Waldhauser, F., & Scholz, C. H. (2021). The shear deformation zone and the smoothing of faults with displacement. *Journal of Geophysical Research: Solid Earth*, 126(5), e2020JB020447.
- Powers, P. M., & Jordan, T. H. (2010). Distribution of seismicity across strike-slip faults in California. *Journal of Geophysical Research: Solid Earth*, 115(B5).
- Rodriguez Padilla, A. M., Oskin, M. E., Milliner, C. W., & Plesch, A. (2022). Accrual of widespread rock damage from the 2019 Ridgecrest earthquakes. *Nature Geoscience*, 1-5.
- Tan, Y. J., Waldhauser, F., Ellsworth, W. L., Zhang, M., Zhu, W., Michele, M., ... & Segou, M. (2021). Machine-learning-based high-resolution earthquake catalog reveals how complex fault structures were activated during the 2016–2017 Central Italy sequence. *The Seismic Record*, 1(1), 11-19.

Detailed responses to Reviewers' comments

The reviewers' comments are quoted in black. Our responses follow in blue. Line numbers refer to the revised version of the manuscript with changes highlighted in color. For this .docx document we used the following margins: Top 1.75; Bottom 1,42; Left 1,42; Right 3.5. Different formats can change the line numbers.

Referee n°1, R1

The paper On fault vs. off-fault seismicity: the role of rock vs. fault rheology by Collettini et al., brings a new interpretation of the diffuse seismicity in the Apenines, which does not occur along the main faults and shows higher b-values. They associate its occurrence to the heterogeneous structure of the rock medium, which is by part ductile and by part brittle.

The explanation of the diffuse seismicity by the heterogeneous geology fabric, which is supported by drillings, outcrops and rheology is novel and deserves publication.

The paper is interesting and reasonably well written. However, the first part on Geology and seismicity observation would benefit from some rewording, which also applies to the Discussion as it is not easy to read and understand. I also believe that adding the viewpoint of seismicity evolution in time of focal mechanisms would make the argumentation more robust and transparent. A moderate revision is, to my opinion required, to make the paper acceptable for publication.

We thank R1 for his/her appreciation of the novelty of our findings. His/her constructive criticisms helped us clarifying several sections of the main text. In particular, his/her suggestion to perform an analysis of the time evolution of the daily number of earthquakes and magnitudes also makes the manuscript stronger.

GENERAL POINTS

1) The diffuse seismicity in dolostones and anhydrites could show variety of focal mechanisms than the on-fault earthquakes. The reason is the strongly heterogeneous structure of brittle dolostones encapsuled in ductile anhydrites and also fluid patches. So I believe tha analysis of focal mechanisms would definitely provide verification of the proposed model. However, It would require to extend the study, which is probably limited by length.

Following the suggestion of the reviewer, we have selected the available focal mechanisms ($M > 3.0$) in the area where Triassic Evaporites are located. All these focal mechanisms occur in the downdip hangingwall seismicity, DHwS (Figure below). Unfortunately, the number of focal mechanisms is limited to only 18 events. Most of the focal mechanisms show extensional kinematics and NW-SE trending nodal planes, in agreement with the regional stress field. The few strike-slip focal mechanisms show left-lateral kinematics for N-S trending planes and right-lateral kinematics for E-W trending planes, again in agreement with the regional stress field (NW-SE trending extensional faults and N-S trending transfer faults inherited from the compressional phase, see for example some details in Collettini et al., 2005). For what is worth with such limited number of events, these data suggest that earthquakes with $M > 3.0$, and therefore occurring on planes with dimensions larger than about 400 m, are controlled by the regional stress field. However, we stress here that 18 earthquakes are not enough to build a solid statistic.

Further insights can be gathered from field data of brittle faults/fractures with dimensions of about 10-100 m in Triassic Evaporites showing extensional kinematics, with an orientation approximately consistent with the extensional regional stress field and some scattering in the strike of the fault/fracture planes (De Paola et al., 2008, see figure below). Ruptures with dimensions of about 10-100 m are consistent with earthquake magnitudes in the range 0-2. For such small magnitudes, traditional focal mechanisms retrieved from P-phase polarities are not available. However, recent unpublished data of 65,000 focal mechanisms for the earthquakes of the sequence with magnitude down to $M = 0$ and obtained using a machine learning approach (Men-Andrin personal communication 2022), show a significant predominance of extensional focal mechanisms with nodal planes showing a similar scattering in strikes to the field ones.

To conclude this point, the analysis on the focal mechanisms requires a statistically more robust dataset, which is not available at the moment. However, preliminary analysis of low-magnitude earthquake (M 0-2) within the TE and small fault/fracture systems observed in the field are in apparent accord with the few larger events focal mechanisms ($M > 3$), all showing extensional kinematics, with an orientation approximately consistent with the regional stress field. We feel that an extensive analysis of focal mechanisms, required to validate these preliminary observations, is beyond the scope of this work. However, we believe that the model proposed in this work, is verified, and consolidated via the analysis of time evolution of the seismicity (i.e., daily number of earthquakes and magnitudes) we developed following the suggestion of R1 (see point 4).

2) It is hard to understand the space relations between seismic clusters, faults and geological units in Fig. 2. I also think that the Supplementary Fig. 1 is unnecessarily detailed, compared to its sake - to show the occurrence of on- and off-fault seismicity.

To address these issues, we have:

1. improved the description of the seismicity at lines (114-126 and in the new Supplementary Note 2).
2. modified figure 2 to better present the space relationship between seismic distribution, faults, and geology.

3. reduced the number of cross sections in Supplementary Fig. 1, which has now become Supplementary Fig. 3.

3) While the seismicity analysis is convincing (but only after repeated reading), I got a bit lost in the geological part distinguishing anhydrites and dolostones and their rheological behaviour. On one hand there is a large number of photos showing the outcrops that illustrate the rock rheology (I think less could suffice, but it I am mainly a seismologist...). On the other hand, I feel that the data, photos, and references provided (too many references which are rather shallow, without any explanations) could allow for different interpretations of the off fault seismicity. Possibly fewer references could be given and focus to some of the references points in the text - this applies to the entire paper.

We have improved the description of the geological dataset by deleting redundant parts of the text, removing unnecessary references, and adding short paragraphs to clarify the key points of the paper (see also our addressing of comments from R3). To make the text more focused we have:

- a) merged the two chapters dealing with structural geology and rock deformation experiments in one single chapter. This chapter addresses rock vs. fault rheology and begins with a brief introduction to tune the reader on its contents (Lines 179-182).
- b) removed from the geological part some redundant text and its references (e.g., lines 183-186), to make the text more focused on the key topics of the paper.
- c) improved the paragraph to better clarify the adopted rheological terminology (brittle vs ductile), which now refers to examples from the field and the lab (lines 194-199).
- d) added a small paragraph to summarize the main message of the Rock vs. fault rheology section (lines 215-220).
- e) added a paragraph dealing with the low-permeability of the anhydrites and field evidence for fluid overpressure (lines 235-241).

To make the paper stand on its own, and rely as little as possible on references from previous work, we have added details about (note that these changes also address some similar points raised by R3):

- a) the background on fault structure, aftershocks, and stick-slip behaviour (lines 33-53).
- b) the description of seismic reflection profiles used to build Fig. 2a (new Supplementary Note 1 and Supplementary Fig. 1 and 2).
- c) the description of the seismological data (lines 114-126 and in the Supplementary note 2), and how they integrate with subsurface geology (see answer to point 2).
- d) the selection criteria for the b-value analysis (lines 147-150, and new Supplementary Note 3 and improved Supplementary Fig. 4-6).
- e) the seismicity occurring in the clusters C1-C3 (160-167 see also answer at point below).
- f) rock vs. fault rheology section.
- g) the main message contained in some of the references, see for example lines 315-321.
- h) the interpretation of seismicity of clusters C1-C3 (lines 292-306).

4) The authors distinguish between on-fault and off-fault seismicity by different b-value. I like this measure for seismicity classification; the b-value estimation looks well based. And this brings me to the fact that similar difference in b-values is observed for mainshock-aftershock sequences and swarms. And on top of that, mainshock and aftershocks usually occur on fault planes, but swarms can also be diffused seismicity. I think the study would gain from incorporating this point of view. By checking other features typical for swarms to see if the off-fault seismicity is indeed missing mainshocks. This would require at least showing the magnitude-time plots. I also remind that the earthquake swarm occurrence is, despite to the

fluid pressure propagation, also associated to the aseismic slip (creep) in a ductile environment, with brittle asperities distributed. And the Appenines diffuse seismicity might be this case.

Following the reviewer’s suggestion, we have performed the analysis of the time evolution of the daily number of earthquakes and magnitudes for both on-fault and distributed seismicity. The clusters C1-C3 show multiple increase and decrease of seismicity rate in agreement with a typical swarm-like activity. The plots presented below, integrated in the new version of figure 3, nicely show the swarm-like activity of the three clusters.

Differently, for on-fault and the DHWS (distributed down-dip hangingwall seismicity), the seismicity rate after the Norcia mainshock follows Omori’s law. The Omori’s type behaviour for the DHWS within the Triassic Evaporites is consistent with seismicity triggered by the mainshock shear stress increase. We have also included these new data in the revised version of the manuscript (Fig. 3).

Explanation for these new analyses and data is reported in the revised version of the manuscript at lines 160-167.

PARTICULAR COMMENTS

75: Reference fo Miller et al (2004) appears not suitable here when speaking about seismic sections and geology.

Agreed, we have removed this citation from the manuscript.

90: Even after repeated reading of this sentence and Fig. 2 I did not identify the 6-8 km wide and 4 km thick zone of diffuse seismicity.

We have improved this part adding a better description of seismicity distribution in space and time (lines 114-126), together with an improved version of figure 2.

104: The language is extremely brief and enigmatic - for example I find not acceptable to say ‘but see also 33’. At least the basic message of the reference should be mentioned.

Agreed, we have modified the text to explain the main message of the cited references (lines 315-321).

106 - 110: I did not get if a) and b) cases are somehow related to Fig 3a and 3b; besides the notation DHwS is not mentioned in Fig.3 caption

We have improved text (lines 114-126) and the labeling of Fig. 3 to clarify this point.

119: Referring to stress increase 39, 40 is again too brief - needs to be explained what stress increase and why. Perhaps indicating the Visso Norcia mainshocks in Fig. 2 in relation to the DHwS events could help.

Agreed, following the reviewer suggestion, we have specified the Norcia mainshock in figure 2 and clarified that we refer to shear stress (lines 169).

137: What do you mean by 'elasto-frictional stress drop'? By comparing with the next line I find that probably 'non-zero stress drop would sound clearer'

In the revised version of the manuscript, we have introduced a definition of elasto-frictional behaviour and associated stress-drop at lines 45-53. For the specific point, we also mentioned data documenting elasto-frictional behaviour and stress-drop, Fig. 6a black curves and 6c, (lines 197-199).

188: What mainshock nucleation do you have in mind?

We have now specified at line 258 that we refer to Norcia mainshock.

266: Why is the red off-fault seismicity limited from top exactly by 6.1 km? From the location plot it appears that also the events up to the 5km depth belong to it.

Yes, we agree with that, but for the selection criteria we have to set a geometrical volume and to do that we have used a sort of parallelepiped (cf. new Supplementary Fig. 5). This parallelepiped is of course an approximation that contains most of the seismicity within the TE. In the revision process, we improved the selected volume (to better approximate the DHwS seismicity) and the obtained *b*-value has been confirmed.

280: It would be very useful to see the time history of seismicity - magnitude vs. time - ideally for the on-fault and off-fault seismicity. This would help to see the time relation of the main earthquakes and also how the off-fault seismicity is related to the on-fault mainshock.

Yes, we agree with that and have performed the analyses suggested by the reviewer (see details provided at point 4).

References

Collettini C., Chiaraluce L., Pucci S., Barchi M. R. and Cocco, M. (2005). Looking at fault reactivation matching structural geology and seismology. *J. Structural Geology*, ISSN: 0191-8141.

De Paola N, Collettini C., Faulkner D.R. and Trippetta F. (2008). Fault zone architecture and deformation processes within evaporitic rocks in the upper crust. *Tectonics*, vol. 27, ISSN: 0278-7407.

Miller, S. A., Collettini, C., Chiaraluce, L., Cocco, M, Barchi, M. R. & Kaus, B. Aftershocks 390 driven by a high pressure CO₂ source at depth. *Nature* 427, 724–727 (2004).

Referee n° 2

This paper explores how on- and off-fault seismicity might be controlled by the rheological behavior of the host rock in comparison to the frictional behavior along the fault. Using the Norica earthquake sequence, the authors map seismicity on and off fault, detail structures in the same rock types in the field, and run a set of strength and frictional experiments to compare on and off fault behavior. The authors conclude that diffuse off-fault seismicity is driven by ductile deformation off-fault that is triggered by on-fault seismicity and possible fluid pressure changes. Although it seems pretty likely that both ductile and brittle deformation is taking place here, as evidenced by the abundant examples in outcrop, the tie to the seismicity is less convincing. I think there are a lot of interesting things to unpack here, but some clarity and perhaps some caveats should be added to the paper.

We thank Reviewer 2 (R2) for his/her positive comments about the content of the paper. Following R2 suggestions, we revised the text of the manuscript and addressed his/her constructive criticisms. We believe that the clarity of the text is now much improved, and that the interpretations of the seismicity dataset is more robust.

Below are the main issues that need to be addressed.

1a) It is not convincing from Figure 2 that the diffuse seismicity is mostly limited to the Triassic evaporites (TE). The off-fault seismicity in the footwall containing the clusters studied in more detail (C1-C3) looks like it might be on vertical structure that goes through both TE and the carbonates. I would suggest looking at a time lapse of seismicity to determine whether this is actually an aseismic event that actually starts in the TE and then migrates into the carbonates. R2 is correct, distributed seismicity also occurs in the carbonates, which are located above and below the TE. We added a statement at lines (173-177) to reinforce this observation.

As suggested by R2, we expanded our analyses and investigated the time-space evolution of seismicity in the TE and in the carbonates located below and above the TE.

The seismicity in the carbonates below clusters C2-C3 is not very clustered in space (Fig. 1A and B below) and in time, but rather occurs during the entire year of the catalogue (Fig. 1C). On the contrary, most of the seismicity of clusters C1-C3 in the TE is occurring in short time-intervals (< 2 months, new Fig. 3f-h main text). Further, the b-values for the carbonate-seismicity below clusters C2-C3 is significantly lower (1.43 in Fig. 1D below) than the b-values of clusters C1-C3 (1.66-1.81; see main text Fig. 3c). These observations further emphasize the difference in rheology and slip behaviour between the seismicity occurring in the carbonates (below the clusters C2-C3) and in the TE (clusters C1-C3).

Figure 1: seismicity on carbonates at depth of about 4-6 km.

The seismicity in carbonates above C2-C3 occurs approximately during the same time-interval of C2-C3 events (before Norcia mainshock). Hence, we performed further analyses on the dataset to test the interesting point raised by the referee, i.e. *“to determine whether this is actually an aseismic event that actually starts in the TE and then migrates into the carbonates”*. We have plotted in space and time the seismicity occurring in C2-C3 and in the carbonates above; these events are highlighted in purple in Figure 2a below and plotted in map view (Fig. 2b) and cross-sections (Fig. 2c and 2d). The grey circles around the hypocenters in Fig. 2b-d (only clearly visible for $M > 1.5$) represent the rupture dimensions, estimated assuming a circular rupture and a constant stress-drop of 3 MPa (e.g. Kanamori and Brodsky, 2004). The cross sections in Fig. c and d show that most of the seismicity within the carbonates occurs in September 2016 (blue colors), whereas the seismicity within the TE mostly occurs in October with some deeper events (3-4 km) occurring in September. Therefore, the lag observed between events in carbonates and TE suggests that the hypothesis of a possible aseismic event starting in TE and migrating into the carbonates is unlikely. At the same time, we note an evolution in time of the seismicity within the TE volumes (see blue and yellow-red events in Fig. 2c-d). These observations are certainly very interesting and worth further investigations, which we feel it is beyond the scope of this manuscript.

Figure 2: seismicity on carbonates above C2 and C3 and on C2-C3.

To summarise, our preliminary analysis of the time-space evolution of seismicity shows that there is no clear link between the seismicity in clusters C1-C3 and the events in the carbonate units located below and above C1-C3. We feel that an in-depth investigation of such differences is beyond the scope of the manuscript. Hence, we would prefer not including these preliminary results and observations in the paper and therefore we added this part in the Supplementary Note 4 and Supplementary Fig. 7.

1b) Furthermore, the authors point out that there is a lot of seismicity in the TE where the fault soles into a detachment, but is that not because there is a detachment there? Or are they arguing that a detachment is not required here? This is presented as an alternative hypothesis to a reactivated thrust fault, but is it not likely that an old thrust might be located within the anhydrite layer?

It is unfortunate that the lack of clarity of the text may have misled the reviewer about whether a detachment fault affects the location of distributed seismicity in the TE.

Detachment faults are usually defined as regional low-angle normal faults cutting through the entire seismogenic layer and accommodating a large amount of extension (e.g., Wernicke 1981; Collettini, 2011). In this work we show that most of the seismicity located at 5-9 km depth is not localized along a detachment fault, but instead occurs as up to 4 km thick imbricated seismicity bands (see Figure 3 below from sections 7 of Supplementary Fig. 3 for our interpretation). The base of the imbricated bands coincides with the top of the basement (dashed white lines) that is affected by compressional steps, i.e., thrusts rooted into the basement, formed during the Late Miocene-Early Pliocene compressional tectonic phase (Barchi et al., 2021). In our interpretation, the seismicity bands at 5-9 km of depth are hosted within Triassic Evaporites resting on top of the basement (see picture below).

Figure 3: interpretation of the seismicity distribution along cross section 7 of Supplementary Fig. 3.

Differently SE of the Norcia mainshock, at depth of about 9-12 km a gently east-dipping structure is present (red arrows in Fig. 2 and sections 4 and 5 of Supplementary Fig. 3). This structure, highlighted by continuous seismicity alignments, has been interpreted by previous authors as an extensional detachment already reported in the literature (Chiaraluce et al., 2017; Michele et al., 2020; Waldhauser et al., 2021). This deeper (9-12 km) seismicity within the basement is distinct from the distributed seismicity in the imbricated bands of TE and is not considered in our analysis.

In the revised version of the manuscript, to discriminate between seismicity on a possible detachment and DHWS, we have added: a) the description of the detachment within the seismological data (lines 134-138); b) the discussion of the manuscript at lines 264-279; and in c) the new Supplementary Note 2.

2) The authors show that the B values are lower on fault than off fault, which they attribute to off fault seismicity being driven by ductile failure. However b-value changes are terribly non-unique and an increase in b-values has been attributed to fluid pressure, fault size distribution, stress, and more. Could aseismic ductile deformation create a higher b-value? Sure, but we cannot prove that is what's happening here.

We agree with R2 that a large number of factors can influence b -values. However, our interpretation are based on evidence derived from the integration of: a) field observations (e.g., large number of small faults within the TE outcrops); b) experimental datasets (showing that low permeability and high pore pressure can control ductile-brittle rheology of TE); c) borehole measurements (e.g., high-fluid pressures documented within the TE in deep boreholes and inferred from hydrofracture systems); d) seismic tomography (e.g., high V_p/V_s anomalies observed during the sequence). Furthermore, significantly lower b -values estimated within carbonate lithologies provide additional independent support to the interpretation that distributed ductile deformation and compartmentalized fluid overpressures trapped within TE are the likely cause of the observed high b -values in the TE.

To address the reviewer comments, we have expanded the discussion clarifying this point. In a sub-paragraph, we acknowledge that different factors can influence b -values (lines 315-322). Then, we propose our interpretation (lines 322-301) that ductile-brittle rheology in concert with fluid overpressure can explain the observed b -values in the TE. In the revised version, more data have been added and discussed at lines (299-306) to further support evidence for fluid overpressure in the study area.

3) If there is a detachment fault that the more steeply dipping fault soles into, why is the detachment not included in the on-fault seismicity (Supplementary Figure 2)? This including some amount of that seismicity as being on-fault might change the b -values.

This point has been extensively addressed at point 1b. In brief, our interpretation is that most of the seismicity located at 5-9 km depth occurs within imbricated bands of TE, which are located above the basement (see Figure 3 of this note). The detachment highlighted in supplementary figure 2 of the previous version of the manuscript (now is in Fig. 2 of the paper and in Supplementary Fig. 3) is located at depth between 9-12 km, i.e. at depths not considered in our analysis.

In addition, following the suggestions of R3, we have provided a more precise definition and applied a more rigorous method to discriminate between on-fault vs. off-fault seismicity. For on-fault seismicity we refer to aftershocks located within the fault structure that is activated by the mainshock and, therefore, this is another reason for not including the deeper detachment seismicity. We think that the actions we have taken to follow the suggestions of R3 (see below for details) will further clarify this point raised by R2.

References

Barchi, M.R., Minelli, G., Piali, G., 1998. The CROP 03 profile: a synthesis of results on deep structures of the Northern Apennines. *Memorie della Società Geologica Italiana* 52, 383–400.

Barchi, M. R., Carboni, G., Michele, M., Ercoli, M., Giorgetti, C., Porreca, M., Azzaro, S. & Chiaraluce, L. The influence of subsurface geology on the distribution of earthquakes during the 2016–2017 Central Italy seismic sequence. *Tectonophysics* 807, 228797 (2021).

Chiaraluce L. et al. The 2016 Central Italy Seismic Sequence: A First Look at the Mainshocks, Aftershocks, and Source Models. *Seism. Res. Lett.* 88, 757-771 (2017).

Collettini, C. (2011). The mechanical paradox of low-angle normal faults: Current understanding and open questions. *Tectonophysics*, 510, 253–268, ISSN: 0040-1951.

Kanamori, H. & Brodsky, E. E. The Physics of Earthquakes. *Rep. Prog. Phys.* 67, (2004).

Improta, L. et al. Multi-segment rupture of the 2016 Amatrice-Visso-Norcia seismic sequence (central Italy) constrained by the first high-quality catalog of early Aftershocks. *Scientific Reports* 373 9, 6921 (2019).

Michele, M., Chiaraluce, L., Di Stefano, R. & Waldhauser, F. Fine-scale structure of the 2016–2017 Central Italy seismic sequence from data recorded at the Italian National Network. *Journal of Geophysical Research: Solid Earth* 125, e2019JB018440 (2020).

Waldhauser, F., Michele, M., Chiaraluce, L., Di Stefano, R. & Schaff, D. P. Fault planes, fault zone structure and detachment fragmentation resolved with high-precision aftershock locations of the 2016-2017 central Italy sequence. *Geophysical Research Letters* 48, e2021GL092918 (2021).

Wernicke, B., 1981. Low angle normal faults in the Basin and Range Province: Nappe tectonics in an extending orogene. *Nature* 291, 645–648.

Referee n° 3

C. Collettini et al. propose to combine geological and seismological data to investigate the possible differences in seismicity distribution and associated laws (e.g., Gutenberg-Richter) following the Norcia-Amatrice earthquake sequences that occurred in 2016-2017. This original approach allows the authors to have a detailed picture of the fault structure thanks to the high-precision earthquake relocation catalog available in this area, and the link with the local geology which presents both brittle (carbonates) and ductile (evaporites, anhydrites) rheologies. They find that localized “on-fault” seismicity is mainly related to the main rupture of the fault plane in carbonates, responsible for the earthquakes. On the other hand, diffuse “off-fault” seismicity exhibits “higher” b-values than on-fault seismicity, and would be related to the rheological ductile behavior of evaporites.

While I found this study very interesting, associated with very nice data, I express below some major concerns that the authors should consider. They are related to different aspects of the study such as the definition of on-fault and off-fault seismicity or the lack of detailed descriptions of the methods (in particular because a large part of the work used in this paper has been published in previous studies). There are several typos in the text and figures and I suggest few improvements to add some clarity between figures.

We thank the reviewer for his/her appreciation of our work, and for providing some constructive criticisms to our manuscript. Below is a detailed account of our addressing of R3 comments and suggestions. We believe that the revised version of the paper has been significantly improved in terms of description of the methods and clarity of the text.

1) *Methods and definition of on-fault vs off-fault seismicity:*

I understand that the authors have to define some rules to select the seismicity. But I have some difficulties with their definition of on-fault and off-fault seismicity. Usually off-fault seismicity is the seismicity situated away from the fault plane (ie. perpendicularly to the fault strike and dip). And some papers have studied this seismicity distribution during interseismic and early postseismic periods, sometimes in direct link with geological observations (e.g., Powers and Jordan 2010; Hauksson, 2010; Perrin et al., 2021; Rodriguez Padilla et al., 2022). I think it is important that the authors better describe in the text what they call on-fault vs off-fault zones to avoid any confusion with existing studies that are using the same words to describe different things (that also varies from tectonics to seismology...!). Maybe an alternate wording of what the authors measured would be “deformation in localized vs distributed fault sections”.

We thank R3 for raising this important point, and we agree that our choice of terminology to discriminate between “on- and off-fault seismicity” may induce some confusion, especially in the context of the existing literature.

For “on-fault seismicity”, we intended the seismicity occurring along the major active faults of the Apennines where the mainshock nucleates. For this on-fault seismicity we mean all the aftershocks occurring in the fault core, the damage zone and possibly extending through the fault loading medium or the *shear deformation zone sensu Perrin et al., 2021*. For “off-fault seismicity”, we intended aftershock distribution within volumes of the crust not including major faults hosting mainshocks.

In the revised version of the manuscript, we have made the following changes to clarify the definitions of on- and off-fault seismicity:

a) Expanded the introduction to present the “geological” (Chester et al., 1993; Mitchell and Faulkner, 2009; Faulkner et al., 2010, Savage and Brodsky, 2011) and “seismological” (Powers and Jordan 2010; Hauksson, 2010; Perrin et al., 2021) views of fault structure. In the revised

text, we clarify that the “seismological view” of the fault is mainly highlighted by aftershock distribution, in accord with selected previous studies (Powers and Jordan 2010; Hauksson, 2010; Perrin et al., 2021).

b) Added some text upfront at lines 33-45 to explain the link between fault structure and fault rheology (e.g., the elasto-frictional behaviour).

c) Stated the revised adopted terminology at the end of the introduction (lines 76-80). Following R3 suggestions, in the revised text, we have adopted the definition of **on-fault vs distributed seismicity**. We have clarified (lines 76-80) that with “on-fault” seismicity we refer to aftershocks located within the fault structure that is activated by the mainshock. We have explained that this fault structure contains the fault core, damage zone and at least part of the fault loading medium. We preferred to avoid the term “localized” because its scale dependency can create confusion in the context of our multiscale dataset. We have also clarified that for “distributed seismicity” we refer to abundant aftershock occurrence within volumes of the crust not including major faults hosting mainshocks.

2) Here the off-fault seismicity considered by the authors is in the continuity of the fault plane along dip, which to me includes both parts of on-fault deformation rooted in a decollement layer and some off-fault deformation. The seismic profiles show that normal fault and thrust are merging at depth where the diffuse seismicity is observed. It seems hard to me to think that these fault planes are considered here as a “off-fault” zone selected by the authors (supp. Fig. 2). At least, the authors are not decorrelating both signatures in their selection.

We think that the lack of clarity in the text about our definition of on-fault seismicity and the methods used to select seismicity for the *b*-values analysis may have induced some confusion. Aftershock distribution and co-seismic slip of the Norcia M 6.5 mainshock do not show a clear image of the prosecution of the fault that hosted the Norcia mainshock (*earthquake fault*) below hypocentral depth (> 6.1 km). Similarly, seismic reflection profiles do not provide unambiguous tracing of the fault trajectory at depths > 6.1 km (*geological fault*). Following our analyses, we interpret that the Norcia earthquake fault at depth > 6.1 km terminates in a zone made of structural and lithological heterogeneities in the TE. It is for this reason that the base of the on-fault seismicity has been set at 6.1 km.

To clarify our interpretations we have revised the manuscript by:

a) adding some text to comment about the prosecution at depth of the main fault that hosted the Norcia earthquake (main text lines 147-150);

b) presenting new supporting data for the adopted selection new Supplementary Note 3 and Supplementary Fig. 4.

3) In the same way, it seems that the width of the on-fault box is about 1km (or 2*500m each side from the fault plane, Supp. Fig 2), which is a significant volume where on-fault and off-fault processes are already occurring and could be discussed. For example, Perrin et al. (2021) have shown that diffuse off-fault deformation could occur up to 4-5 km away from the fault plane in case of immature fault cases (and the normal faults in the Apennines are likely immature).

In the revised version of the manuscript, with the definition of the meaning of on-fault seismicity we have clarified that within the 1 km box we are considering all the processes occurring within the fault core, damage zone and the fault loading medium or the zone of shear deformation (Perrin et al., 2021).

R3 also raises a very interesting and stimulating point about fault thickness and maturity referring to the work of Perin et al. (2021). We think that Perrin et al. (2021) is a great work,

linking geological and aftershock observations on active faults. We thank R3 for bringing it to our attention, as we were not aware about this work. We feel that the analysis of the relationships between fault thickness and cumulative displacement is beyond the scope of the manuscript. However, we would still like to present in the following some key geological and seismological observations and comment, at a speculative level, on their relevance to the topic of fault thickness and maturity.

Perrin et al. (2021) find that for strike-slip faults the width of the shear deformation zone, as defined by aftershocks distribution, decreases with fault maturity (cumulative fault displacement, length, initiation age and slip rate). In the Apennines, active normal faults have similar age < 2 Ma, and displacement, 1-2 km (e.g. Roberts and Michetti, 2004). This is in part controlled by the extensional environment and the shallow ≈ 12 km brittle-ductile transition. Due to these environmental constraints, normal faults dipping at 45-60° in a relatively thin (12 km) seismogenic layer cannot accommodate large displacements. Nevertheless, some differences are still observed between seismic sequences. For example, the Norcia 2016 Mw 6.5 mainshock occurred on the M. Vettore fault with a maximum cumulative displacement of 1.3 km (Porreca et al., 2020) and an aftershock width of about 1.74 km (Waldhauser et al., AGU, 2020; Waldhauser et al., 2021 GRL). Whereas the L'Aquila 2009 Mw 6.1 mainshock occurred on the Paganica fault with a maximum cumulative displacement of 0.6-0.8 km (Pucci S. personal communication and Blumetti et al., 2013) and an average aftershock width of about 0.7 km (Valoroso et al., 2014). Notably, the Norcia earthquake occurred in the Umbria-Marche domain (cf. figure below), which is a stratigraphically and mechanically complex multilayer due to the presence of Triassic evaporites and some marly lithologies (i.e. formations) within the Cretaceous-Jurassic sediments succession. We speculate that mechanical heterogeneities of the TE may contribute to creating a thicker and more complex or rough fault structure. Differently, L'Aquila mainshock occurred in the Gran Sasso domain, which is characterized by more competent and relatively homogeneous Cretaceous-Triassic-Paleogene succession of shallow water carbonates. We speculate here that the absence of TE, and more generally of lithological heterogeneities, may produce a thinner and less rough fault structure. To summarise, we speculate that significant differences in lithology between the Northern (Umbria-Marche domain) vs. Central (Gran Sasso domain) Apennines may explain the observed differences in aftershock width for faults with similar maturity (age and displacement).

We reiterate here that while the analysis of the relationships between fault thickness and displacement is beyond the scope of the manuscript, future research should investigate the role that fault maturity (sensu Perrin et al., 2021) and lithological heterogeneities play in controlling the width of the zone of shear deformation in the Apennines.

4) *New contributions from this paper vs published studies and data availability statement:* In short, the paper should stand on its own.

This point was also raised by R1. In the revised version of the manuscript, we provided more information from previous studies without replicating them. More specifically, we have improved the:

- introduction, to set the background on fault structure, aftershocks, and stick-slip behaviour.
- description of seismic reflection profiles used to build Fig. 2a (new Supplementary Note 1 and Supplementary Fig. 1 and 2).
- description of the seismological data at lines (114-128 and in the new Supp. Note 2)
- description of the selection criteria used for the b-value analysis at lines 147-150, and details in the new Supplementary Note 3 and improved Supplementary Fig. 4-6.
- the analysis of the time evolution of the daily number of earthquakes and magnitudes for the seismicity occurring in the clusters C1-C3 (lines 160-167 and new data in Fig. 3).
- the rock vs. fault rheology section.
- the discussion to clarify the interpretation of the seismicity of C1-C3 (lines 292-308).
- the method section (lines 345-362).
- the data availability statement (lines 418-423).

5) Data availability statement

The data availability statement is not conformed to current publication policies: the supplementary material of this paper is actually not providing data but figures. The seismicity catalog used should be placed in an open archive or the author should cite the web address and associated references (zenodo link in Tan et al, 2021 paper). Same for the rheological tests, etc.

We have improved the data availability section. Here as suggested by the referee we have cited the web address and the associated references for seismological and frictional data. For the triaxial tests on anhydrites we have created an *ad-hoc* data repository. For the b-values analyses we have released the Python Jupyter notebook on github with related link in the Code availability (lines 426).

We also understand from the referee that we could do more for the Supplementary Information. Therefore, in the revised version we have added extensive text (Supplementary notes 1-4) and figures (Supplementary Fig. 1-4 and 7) to provide the reader with more details.

6) Interpretations

The idea that the rheology is impacting the seismicity location and frequency is really interesting. But I would go further in the comparison between off-fault and on-fault deformation in different rheological layers (see figure below).

Figure presented by R3

Following the suggestion of R3, we performed an exploratory analysis of the b -value for the off-fault seismicity. In particular, we calculated b -values for the seismicity occurring in the carbonates between C2-C3 and C1 clusters ($b = 1.43$; blue rectangle in the figure below) and for the carbonates above C2-C3 ($b = 1.30$; red rectangle). These preliminary results are surprisingly interesting showing that b -values in carbonates are significantly lower than those estimated for the C1-C3 clusters in the TE (b in the range 1.66-1.81). We interpret these results as further evidence of the different rheology of the carbonates (more brittle) compared to the TE (more ductile). In addition, two more potential clusters have been analyzed within deep TE. These clusters are C4 and C5 in the figure below, (although C5 has not the typical subvertical geometry of the other clusters), which are characterized by higher values of b equal to 1.66 (C4) and 1.50 (C5). Finally, the east-dipping, shallow and antithetic normal fault (blue ellipse in the figure below) shows $b = 1.35$.

We did not perform the separate calculation of b -values for on-fault seismicity in carbonates vs. TE, as suggested by R3 (green solid ellipse in the figure provided by the referee) for two main reasons. A) Field geology shows that the structure of normal faults with relatively large

displacement (several hundreds of meters or more), in both carbonates and Triassic Evaporites is the typical structure of brittle faults. Field observations show the development of a fault core affected by cataclastic processes with grain-size reduction and localization (Figure 5), and mechanical data show elasto-frictional and stick-slip behaviour (Figure 6b). Therefore, we think that from the Earth surface down to hypocentral depth there are no significant structural or frictional differences along the fault. B) The on-fault seismicity in carbonates highlighted by the green solid ellipse in the figure presented by R3 is mostly located along the contact between carbonates and TE. Hence, it includes the TE that are present along the footwall of the fault.

For what concerns the red ellipse below the hypocentral depth in the figure presented by R3, we have already discussed in detail above (point 2) that seismological and geological evidence in support of the prosecution at depth of the main fault that hosted the Norcia earthquake are ambiguous. Therefore, we did not consider this seismicity as on-fault seismicity.

In conclusions, while these preliminary results are certainly interesting, we would still like to keep the focus of our paper on the distributed seismicity occurring within the TE. An in-depth discussion about the main differences about carbonates and TE will have to be based on speculative interpretations. Unfortunately, we do not have for the carbonates the same extensive, integrated geological and mechanical dataset that we have available for the TE. Hence, we would prefer to limit our analysis to distributed seismicity within Triassic Evaporites.

Line-by-line comments and corrections:

I. 26: stick slip -> stick slip
Corrected.

I. 47: kilometres -> kilometers

Corrected.

I. 46-47: Well, the seismicity is off the main San Jacinto fault but orthogonal directions might be aligned with secondary faults and thus the seismicity be considered as “on-secondary-fault”. The mix between different scales of observations might be confusing for the reader, especially in the definition of diffuse off-fault deformation. In Ross et al., (2017), orthogonal faults are seen at the surface in the SW of San Jacinto but not in the NW, but it doesn’t mean that faults don’t exist at depth in the NW. In map view, the aligned seismicity patterns are not so diffuse taken individually as secondary faults. This case with orthogonal directions is pretty specific (like Supertition Hills/Elsinore earthquakes, Tottori, etc). Maybe this section needs to better explain more generally the different type of off-fault seismicity with more references (see major comment on the definition of on-fault and off-fault seismicity, damage zones, etc). This point has been addressed in the introduction (lines 33-53 and 76-80) providing a less ambiguous definition of on- and off-fault deformation. We have also removed “off-fault” (lines 62) in the text dealing with the San Jacinto fault (Ross et al., 2017).

Fig1: wrong labels for 1c (b labeled twice). Corrected.

- Do you have the possibility to have a better resolved fault map? or at least add symbols to show fault dip and slip motion.

Yes, in the revised version we added fault dip and kinematics.

- ruptures in light blue are barely visible on figure 1a

OK, we have changed the color of the surface breaks.

- Fig. 1b and 1c: Looking at Barchi et al. (2021) and Ercoli et al. (2020), it seems that the two seismic profiles are actually continuous. Why did you cut them in two pieces? It would be better to have them in a single figure to easily compare with Figure 2.

The seismic profiles are not continuous but partially overlap, and this is the reason why we put two different images of the profiles. We have now provided a comprehensive description of the interpretation of the seismic reflection profiles in the new Supplementary note 1 and Supplementary Fig. 1 and 2.

- Also, there are other east-dipping faults that you didn’t report and which could be associated with the cluster situated in the TE between 1-3 km depth between the Mt Vettore fault and the Norcia fault. This cluster is visible in the early times after the earthquake (figure 2b). That would be interesting to also analyze this “on-fault” secondary seismicity in TE to compare with other GR distribution (see also my major comment).

Yes, in seismic reflection profiles several east-dipping normal faults antithetic to the Vettore fault are present (Porreca et al., 2018, Fig. 7; Ercoli et al., 2020 Fig. 4 & 9). Ercoli et al., 2021 describes the Vettore fault as the major structure of the area that is associated to a set of small displacement normal faults both synthetic and antithetic (the figure below is a detail of Figure 7e from Ercoli et a., 2020). These faults in seismic profiles appear as high-angle seismic discontinuities that cross-cut the gently W-dipping reflectors.

Our geological cross-section of figure 2 is based on Barchi et al., 2021. This geological cross-section builds on seismic reflection profiles crossing the area, but the strike is different since to better capture the relationship between subsurface geology and seismicity it was constructed perpendicular to the strike of the activated Vettore fault (cf. for example figure 1 of the present work).

That said, following R3's point we have revised the seismic profiles of the area and decided to insert the trace of the fault in the geological cross section of figure 2a. The description of the normal fault systems in seismic profiles is now present in the new Supplementary Note 1 and Supplementary Fig 2.

- please specify in the caption that TE = pink dashed line and phB = yellow arrows. Why are you not using arrows everywhere (which would allow the reader to see the different reflectors)?
 Yes, in the revised version of the manuscript, we have now used arrows everywhere to better see the reflectors. We also properly labeled these arrows.

- Barchi et al. (2021) and Ercoli et al. (2020) describe the TE reflector as a thrust. Why do you report it here as the top of evaporites? They also show in the eastern part of fig.2c that the reflectors above and below the thrust are actually carbonates so it is contradictory with the location of evaporites you are highlighting. How do you reconcile this with the geological interpretation in Fig.2?

Thanks for spotting this mistake due to a typo in the label of our figure, where the thrust was mistakenly labelled as TE. Yes, R3 is also right about the fact that in Barchi et al. (2021) the reflectors above and below the thrust are carbonates (bottom figure left). The carbonates below the thrust have been labelled wrongly in Barchi et al., (2021). The original interpretation of Porreca et al., 2018 (bottom figure right) is consistent with our Fig. 2a.

As said before, we have revised the seismic profiles and provided a comprehensive explanation for surface and subsurface geology in the Supplementary note 1 and Supplementary Fig. 1 and 2. We have also changed the main text at lines 96-101.

How confident are you in the stratification on which you depend a lot in your following analysis, especially for cluster C2 and C3?

We are quite confident since in the footwall of the Sibillini thrust located above C2-C3, the top of the carbonates is present and beneath the thrust 2 km of carbonates and 2 km of TE form the back limb of the Sarnano anticline. We have specified that in Supplementary Note 1 (lines 52-58) and in Supplementary Fig. 2.

Fig1 vs Fig 2: Please stay Consistent between your figures: the color of normal and reverse faults are inverted between Fig. 1 and 2. Please stick with one color code (normal= red ; reverse = blue).

Thanks for spotting this erroneous color code. We have changed the colors of thrusts and normal faults in figure 2a as requested by R3.

Fig. 2:

- c label in the figure is surrounded by black arrows.

We fixed this. Thank you.

- in the figure caption, two 'b' labels are described and no 'c'.

We changed b in c. Thank you.

- specify that the red star is the Norcia mainshock.

We have fixed this as requested by R3.

- the use of i), ii) is confusing in a figure caption. Put this detail in the main text.

Yes, we have inserted the details in the main text at lines 114-126 and changed the caption.

- the seismicity seems cut above a depth of 0-0.2 km, which might depend on the method/constraints used for the relocations. But the topography is quite significative. Do you really expect no seismicity at shallow depth? or do you think you would need to refine the relocations/catalog? or consider a shift of the seismicity upward closer to the actual topography? Please discuss.

Most of the seismological catalogues for the Central Italy seismic sequence (Michele et al., 2020; Tan et al., 2021; Waldhauser et al., 2021), have a cut of the seismicity at about 0 km. Some earthquakes at crustal depths above 0 km (asl) are likely, in particular in areas where the topography is high. However, we feel that a discussion about the shift of the seismicity upward in areas with high topography is beyond the scope of this paper, as it would require a refinement of the seismic catalogue of the region.

l. 88: off-faults seismicity -> off-fault seismicity ?

Thanks for spotting this typo. Following R3 suggestions, we changed "off-fault" in "distributed seismicity" in the revised version of the manuscript, so this correction is not required anymore.

l. 88-89: figures are only showing cross sections (vertical axis = depth) so using both terms "wide" and "thick" is confusing. Please add a xlabel in your plots (for example "Across-strike distance") and use the corresponding term in your text for clarity. Also from your description, the diffuse seismicity seems to be rather 2 km thick, so please use depth range to spot exactly what you are describing.

We have changed figure 2 to better show the area of distributed seismicity. Now the area is surrounded by a yellow dashed line and is labeled DHWS, i.e., downdip hangingwall seismicity.

I. 102-103: b-value is also sensitive to other parameters such as depth (but it is debated; e.g., El-Isa and Eaton, 2014) or fault segmentation in complex earthquake sequences like it is the case in your study area. In a preprint Herrmann et al are studying in detail this aspect (https://assets.researchsquare.com/files/rs-1210699/v1_covered.pdf?c=1641579587). I know it is not published (yet), but it might be good to anticipate and discuss this a bit more... Thanks for this suggestion. In the revised version we provided further details on the other parameters the b-value is sensitive to (lines 315-321). We have included the reference El-Isa and Eaton, 2014, and mentioned Herrmann et al at lines 321.

I. 105: 0.5 km: at which distance do you consider that you are not “on-fault” anymore? see also my major comments.

We have explained that at line 147 and also emphasized that widening the on-fault at 1 km yields essentially the same results (Lines 157-158).

I. 107-109: I don't understand this criterion. Why considering as off-fault seismicity events situated down-dip in the direct continuity of the fault/aftershock sequence? This is a huge hypothesis, and the data (seismic profiles and geology) are not agreeing with that. See also my major comment.

We think that we have extensively commented on and clarified this point in the answer provided to the major point 2) raised by R3.

I. 111-116: Don't you think that considering both zones as on-fault in carbonates and in evaporites might favor even more your interpretation on the impact of the rheology? Would you obtain the same results by selecting the seismicity in two wide boxes with the same size (i.e., same across strike width)?

Field observations of large displacement faults indicate that large faults in these lithologies are similar. In addition, the TE are a mixed lithology with anhydrites and dolostones, i.e. Mg-rich carbonates, that concentrate along the major normal faults. Within the fault zone the main deformation mechanisms is cataclasis with grain-size reduction and localization along principal slipping surfaces (i.e., Fig. 5 for a major fault within the TE). So, large faults in both carbonates and TE have the typical structure and processes of faults of the elasto-frictional regime (Sibson, 1977; Scholz, 2019). In the lab, the fault rock is characterized by a brittle failure envelope with friction close to the Byerlee's 1978 range, and some velocity weakening behaviour. We have reinforced this point by adding lines 218-220.

I. 123-153: I like this part.

Thanks a lot.

Fig. 4: the dashed red line is hard to distinguish.

OK, we have changed the color of the line from red to green.

Fig.6: label c hardly distinguishable among other strange characters around (problem in the export/conversion of the figures?)

We fixed this.

I. 212: Maybe I missed something, but at this point it seems that you completely forgot to interpret the clusters C1, C2 and C3 situated in TE. They are not diffuse at all but well localized.

More discussions are definitely needed on this point. Is it the case in l. 220-222? If yes it has to be explicitly said and discussed.

We agree with R3 that interpretation on C1-C3 were hidden in the previous version of the manuscript. So, thanks for raising this point. In the revised version, we have expanded this part at lines 292-306 and we think that this significantly improved the presentation of our data.

l. 227: stick sip > stick slip

We corrected this. Thank you.

Fig. 7:

- nice sketch! However, it reflects part of the “reality”. From your Supp. Fig. 2, the seismicity in the TE ductile zone is not only situated in the hanging wall but also in the footwall. Then you measure a behavior in a distributed zone that encompasses both distributed on-fault and off-fault deformations.

We agree with R3, and in the revised version we have modified the figure to make it more consistent with the text and our interpretation.

- the left sketch shows clear small shear fault planes in the anhydrites but not in the dolostones (contrary to the right part). This might be confusing so please add some in the dolostones.

OK, we have added some small faults also in the dolostones. These structures are very clear in the field.

Also, you don't explain the difference in the caption between grey (on-fault seismicity) and red (off-fault seismicity) stars.

We have corrected the caption of figure 7 and explained the differences between the grey and red symbols.

- A suggestion: in the right parts of your figure, put in grey and red the on-fault and off-fault GR distribution to clearly present the link with the left parts of the figure (stars).

Choose a different color for the mainshock otherwise we think that there is a link only with off-fault seismicity and not on-fault seismicity.

OK, we made the suggested changes to the figure, thank you.

l.269: repetition of “defined”

OK, we corrected the mistake.

l. 286: which is? what do you mean by excluded? Herrmann and Marzocchi (2021) show that aftershocks within 4 days after the mainshock present smaller b-values compared to the aftershocks considered in the following month (or more). What time period did you consider and/or remove?

The figure reported below is Figure 6 of the supplementary material of Herrmann et al., (Research Square, <https://orcid.org/0000-0002-2342-1970>). According to the authors, this plot is useful to define the necessary “safety margin” to reduce the bias due to short-term aftershock incompleteness (STAI). According to this analysis, the Norcia mainshock has the strongest influence on STAI, and +2days have to be removed to avoid the influence of STAI on the b-values. For the other mainshocks the influence is limited at +0.8, +0.6 and +0.4 for Amatrice, Visso and Campotosto mainshocks respectively. We have clarified that in the Method section at lines (401-404).

I.287: Estimation => Estimation?

It is estimation, thanks for bringing this out.

I. 288: which is? see also major comments

In the revised version, thanks to the referee's suggestions, we think that we significantly improved the manuscript to make the paper stand on its own. In the specific case we have added more text and formulas at lines 345-362.

Supplementary Figure 1: Caption includes observations labeled twice a) and b) to describe two different things that are not labeled on the figure. This is very confusing. Please change this way to describe your figures.

We have improved the description of the figure including 4 supplementary notes.

References

Barchi, M. R., Carboni, G., Michele, M., Ercoli, M., Giorgetti, C., Porreca, M., Azzaro, S. & Chiaraluce, L. The influence of subsurface geology on the distribution of earthquakes during the 2016–2017 Central Italy seismic sequence. *Tectonophysics* 807, 228797 (2021).

Blumetti, A. M., Guerrieri, L., & Vittori, E. (2013). The primary role of the Paganica-San Demetrio fault system in the seismic landscape of the Middle Aterno Valley basin (Central Apennines). *Quaternary International*, 288, 183-194.

Byerlee, J., 1978. Friction of rocks. *Pure Appl. Geophys.* 116, 615–626.

Chester, F.M., and Logan, J.M., 1986, Implications for mechanical-properties of brittle faults from observations of the Punchbowl fault zone, California: *Pure and Applied Geophysics*, v. 124, p. 79–106, doi:10.1007/BF00875720.

Chiaraluce L. et al. The 2016 Central Italy Seismic Sequence: A First Look at the Mainshocks, Aftershocks, and Source Models. *Seism. Res. Lett.* 88, 757-771 (2017).

Collettini C., De Paola N, Holdsworth R.E. and Barchi M.R (2006). The development and behaviour of low-angle normal faults during Cenozoic asymmetric extension in the Northern Apennines, Italy. *Journal of Structural Geology*, vol. 28; p. 333-352, ISSN: 0191-8141.

El-Isa, Z. H., & Eaton, D. W. (2014). Spatiotemporal variations in the b-value of earthquake magnitude–frequency distributions: Classification and causes. *Tectonophysics*, 615, 1–11.

Ercoli, M., Forte, E., Porreca, M., Carbonell, R., Pauselli, C., Minelli, G. & Barchi, M. R. Using seismic attributes in seismotectonic research: an application to the Norcia Mw = 6.5 earthquake (30 October 2016) in central Italy. *Solid Earth* 11, 329–348 (2020).

Faulkner, D.R., Jackson, C.A.L., Lunn, R.J., Schlische, R.W., Shipton, Z.K., Wibberley, C.A.J., and Withjack, M.O., 2010, A review of recent developments concerning the structure, mechanics and fluid flow properties of fault zones: *Journal of Structural Geology*, v. 32, p. 1557–1575, doi:10.1016/j.jsg.2010.06.009.

Herrmann et al., (Research Square, <https://orcid.org/0000-0002-2342-1970>)

Michele, M., Chiaraluce, L., Di Stefano, R. & Waldhauser, F. Fine- scale structure of the 2016–2017 Central Italy seismic sequence from data recorded at the Italian National Network. *Journal of Geophysical Research: Solid Earth* 125, e2019JB018440 (2020).

Mitchell, T.M., Faulkner, D.R., 2009. The nature and origin of off-fault damage surrounding strike-slip fault zones with a wide range of displacements: A field study from the Atacama fault zone, northern Chile. *Journal of Structural Geology* 31, 802e816.

Perrin, C., Waldhauser, F., & Scholz, C. H. (2021). The shear deformation zone and the smoothing of faults with displacement. *Journal of Geophysical Research: Solid Earth*, 126, e2020JB020447. <https://doi.org/10.1029/2020JB020447>.

Porreca, M., Minelli, G., Ercoli, M., Brobia, A., Mancinelli, P., Cruciani, F., Giorgetti, C. Carboni, C., Mirabella, F., Cavinato, G., Cannata, A., Pauselli, C. & Barchi, M. R. Seismic reflection profiles and subsurface geology of the area interested by the 2016–2017 earthquake sequence (Central Italy). *Tectonics* 37, 1–22 (2018).

Powers, P.M., and Jordan, T.H., 2010, Distribution of seismicity across strike-slip faults in California: *Journal of Geophysical Research*, v. 115, B05305, doi:10.1029/2008JB006234.

Roberts, G.P., Michetti, A.M., 2004. Spatial and temporal variations in growth rates along active normal fault systems: an example from the Lazio-Abruzzo Apennines, central Italy. *J. Struct. Geol.* 26 (2), 339–376. [https://doi.org/10.1016/S0191-8141\(03\)00103-2](https://doi.org/10.1016/S0191-8141(03)00103-2).

Ross, Z. E., Hauksson, E. & Ben-Zion, Y. Abundant off-fault seismicity and orthogonal structures in the San Jacinto fault zone. *Sci. Adv.* 3: e1601946 (2017).

Savage, H.M., and Brodsky, E.E., 2011, Collateral damage: Evolution with displacement of fracture distribution and secondary fault strands in fault damage zones: *Journal of Geophysical Research*, v. 116, B03405, doi:10.1029/2010JB007665.

Scognamiglio, L., et al. Complex fault geometry and rupture dynamics of the MW 6.5, 30 October 2016, central Italy earthquake. *J. Geophys. Res.* 123, 2943–2964 (2018).

Sibson, R. H. Fault rocks and fault mechanisms, *J. Geol. Soc. Lond.* 133, 191–213 (1977).

Scholz, C. H. The mechanics of earthquakes and faulting. Cambridge University Press, (2019).

Tan, Y. J., Waldhauser, F., Ellsworth, W. L., Zhang, M., Zhu, W., Michele, M., Chiaraluce, L., Beroza, G. C. & Segou, M. Machine-Learning-Based High-Resolution Earthquake Catalog Reveals How Complex Fault Structures Were Activated during the 2016–2017 Central Italy Sequence. *The Seismic Record* 1, 11–19 (2021).

Waldhauser et al., 2020; Illuminating the complex fault zone structure of the 2016-2017 Amatrice, Central Italy, earthquake sequence with a high-resolution, high-density earthquake catalog. American Geophysical Union, Fall Meeting 2020, abstract #S041-06.

Waldhauser, F., Michele, M., Chiaraluce, L., Di Stefano, R. & Schaff, D. P. Fault planes, fault zone structure and detachment fragmentation resolved with high- precision aftershock locations of the 2016-2017 central Italy sequence. *Geophysical Research Letters* 48, e2021GL092918 (2021).

REVIEWERS' COMMENTS

Reviewer #1 (Remarks to the Author):

The authors responded sufficiently to all my comments. I like the new event rate and magnitude time plots; it makes the story clearer. And I acknowledge that focal mechanisms do not show difference between the on- and off-fault seismicity, this is not surprising.

Below is a list of minor revisions that are to my opinion needed to make the manuscript acceptable

Main text

Ln 112: Can the seismogenic volume be 'interested'? It sounds to me unusual wording, but I am not a native speaker.

Ln 116: I think the microearthquakes linear cluster span a range from 6 to 4 km depth, not 2 km

Ln 128: TE unit (and other geological units) is no more marked in any of the figures, add the notation back to Fig. 2, because TE is used in many places on the MS.

Ln 135: I think the deep structure occurs at NE, not SE. And the depth is 9-10 km rather than 9-12 km, right?

Ln 156 and below: When comparing the characteristics of the seismicity it turns out that there are two end members: On-fault seismicity (low b , Omori event rate and Mainshock occurrence) and Off-fault seismicity (high b , varying event rate and no Mainshock). And in between the DHWS, which has high b , but Omori event rate and almost mainshock-aftershock character of magnitudes decay. It would be useful to state this clearly in the text, either using a separate table or a special graphics in Fig. 3.

Ln 162: Visso-Norcia mainshock should read as Visso-Norcia mainshocks, right?

Ln 164: What do you mean by reduction of b-value for the DHwS following the Visso-Norcia mainshocks? You give the b-value for the whole sequence, not for the period after the mainshock. And the difference is rather small (0.07), how does it compare with the errors? Please explain in the text

Ln 197: brittle is missing in front of the parentheses

Ln 299: change Fig. 3e to inset in Fig 3e to be clearer

Fig. 2:

- Please indicate in the caption for which period the quake distribution is shown in (b) and the same for (c). Is it the same period as in Suppl. Fig. 3, i.e. 40 days?
- change 'red star' to 'red star and focal mechanism plot'
- the time periods shown would be best indicated directly in the depth section panels
- the SW dipping plane of the Norcia mainshock is not shown in any of the panels; I believe that the beachball is in the map projection, not the depth section

Fig. 3:

- to make it more clear, add titles to each of the graphs a, b, c indicating the time period; increase the fonts of the legend
- in panel d, please mark the mainshocks by the same stars as in Fig. 1.
- in panel d, I can see only one large magnitude event (6), not two as expected
- please be consistent when using Norcia-Visso and Visso-Norcia mainshocks. I believe it is the same...
- consider using a logarithmic y-scale for the seismicity rate - then the inset would not be needed (the exponential Omori decay would change to linear, which does not matter - it is used sometimes)

Supplements

Ln 82-83: I think 1-2 should read as 1-3 because DHwS is visible also in section 3. And what do you mean by SSE or NNW portions? I do not see DHwS in section 4.

Ln 84: I do not see DHwS in sections 6 and 7. Please mark DHwS by yellow polygon similar to Fig. 2.

Supp. Fig.3:

- According to the map view, the section 3 is the same as the section shown in Fig. 2. But the seismicity is bit different, see e.g. the shape of the cluster C1. I think the reason is not the different width shown (2 or 1 km).
- please explain the meaning of the red arrows in the caption

Supp Fig. 4: What is the difference of b) and c). Is c a zoom of b? Please explain in the caption

Tomas Fischer

Reviewer #2 (Remarks to the Author):

I am happy with the revisions that the authors have made and see no further changes needed for publication.

Reviewer #3 (Remarks to the Author):

I thank the authors for their great work on responding to each of my comments. The new supplementary materials are very valuable and help greatly the reader to understand the whole picture! I'm now convinced that this study deserves to be published in Nature Communications. I only have one or two comments that the authors might want to consider before the final proofs.

Congratulations again.

1) I think very interesting that the new figure/analysis provided by the authors in the rebuttal letter (answer to my comment 6:Interpretations) shows similar results i.e. that b-values in carbonates are lower than in TE, even situated away from the fault plane. It would imply that the on-fault behavior

might not be so important compared to the rheology. Thus this is affecting somehow the results of the paper. The authors state that these results are out of the scope and need further investigations. I understand but since the authors already performed some measurements, I would add them in the paper (at least in the supplementary notes) and discuss a little bit further. I remind the authors that this is a transparent peer review process so reviewer's comments and answers will be available online. So to me it would worth it to describe it clearly in the paper/supplements before other people (mis)interpret the results. There is already a small dedicated part in the text and supplements so it won't take too much characters.

2) Very minor comments: please check the GB or US wording of "kilometre" and "hypocentre". I would use "kilometer" and "hypocenter".

Detailed responses to Reviewers' comments

The reviewers' comments are quoted in black. Our responses follow in blue.

Reviewer #1 R1:

The authors responded sufficiently to all my comments. I like the new event rate and magnitude time plots; it makes the story clearer. And I acknowledge that focal mechanisms do not show difference between the on- and off-fault seismicity, this is not surprising.

We thank R1 for his further effort in reviewing our manuscript and for providing with further suggestions to improve the manuscript. Below is a point by point reply to their main comments, including the changes made to the text to address any issue raised.

Main text

Ln 112: Can the seismogenic volume be 'interested'? It sounds to me unusual wording, but I am not a native speaker.

We changed the word to "affected".

Ln 116: I think the microearthquakes linear cluster span a range from 6 to 4 km depth, not 2 km.

Yes, we agree that most of the seismicity is occurring between 3 and 6 km of depth. However, some microseismicity is also present up to 2 km of depth, aligned in continuity with the earthquake plane. This is shown in Fig. 2b and in Supplementary Figure 4b.

Ln 128: TE unit (and other geological units) is no more marked in any of the figures, add the notation back to Fig. 2, because TE is used in many places on the MS.

We have improved this part in: a) the caption of figure 4, 5 and 7. We have also reported Triassic Evaporites within figure 7.

Ln 135: I think the deep structure occurs at NE, not SE. And the depth is 9-10 km rather than 9-12 km, right?

In the cross sections 4 and 5 of Supplementary Fig. 3 it can be seen that the seismicity associated to a possible extensional detachment (red arrows) reaches 12 km. Cross sections 4 and 5 are located SE of the Norcia mainshock.

Ln 156 and below: When comparing the characteristics of the seismicity it turns out that there are two end members: On-fault seismicity (low b , Omori event rate and Mainshock occurrence) and Off-fault seismicity (high b , varying event rate and no Mainshock). And in between the DHwS, which has high b , but Omori event rate and almost mainshock-aftershock character of magnitudes decay. It would be useful to state this clearly in the text, either using a separate table or a special graphics in Fig. 3.

The main message the manuscript seeks to deliver is that within the seismogenic volume the seismicity not only localizes along the major structures hosting the mainshocks, but it is also distributed within volumes of the seismogenic layer. For clarity, we prefer to leave only the distinction between on-fault vs. distributed seismicity. However, in the manuscript (in the paragraphs describing earthquake distribution and frequency magnitude distribution) the differences within distributed seismicity (DHwS vs. clusters) are mentioned.

Ln 162: Visso-Norcia mainshock should read as Visso-Norcia mainshocks, right?

Yes, we have corrected the text.

Ln 164: What do you mean by reduction of b-value for the DHwS following the Visso-Norcia mainshocks? You give the b-value for the whole sequence, not for the period after the mainshock. And the difference is rather small (0.07), how does it compare with the errors? Please explain in the text

We have decided to remove this sentence.

Ln 197: brittle is missing in front of the parentheses

We added brittle.

Ln 299: change Fig. 3e to inset in Fig 3e to be clearer

We see the possible unclarity so, to help the reader, we have labelled Norcia mainshock in the figure to make it clearer that Fig. 3e shows an Omori-like decay for the DHwS aftershocks following Norcia mainshock. The inset compares the microseismicity before the Norcia mainshock for both on fault (grey) and DHwS distributed (red) seismicity.

Fig. 2:

- Please indicate in the caption for which period the quake distribution is shown in (b) and the same for (c). Is it the same period as in Suppl. Fig. 3, i.e. 40 days?

In the caption the period is stated: **b** seismicity from Amatrice (8/24/2016) to Visso (10/26/2016) mainshock and (c) after Norcia (10/30/2016) mainshock.

- change 'red star' to 'red star and focal mechanism plot'

Agreed, we have added moment tensor solution.

- the time periods shown would be best indicated directly in the depth section panels

We prefer to leave the time periods only in the caption to avoid making the figure too crowded and less clear.

- the SW dipping plane of the Norcia mainshock is not shown in any of the panels; I believe that the beachball is in the map projection, not the depth section.

The focal mechanism is in depth-section and it shows one SW-dipping plane in agreement with the on-fault microseismicity.

Fig. 3:

- to make it more clear, add titles to each of the graphs a, b, c indicating the time period; increase the fonts of the legend

Following the reviewer suggestion, we have increased the font of the legend. We prefer to leave the time periods only in the caption to avoid making the figure too crowded and less clear.

- in panel d, please mark the mainshocks by the same stars as in Fig. 1.

Yes, we agree with that and we modify the figure accordingly.

- in panel d, I can see only one large magnitude event (6), not two as expected

In panel d the large magnitude event represents the Norcia earthquake. The other $M \approx 6.0$ event (Visso earthquake) is not contained within the volume of the selected earthquakes for on-fault seismicity. Full details on the methods and selection of events are provided in Supplementary Note 3.

- please be consistent when using Norcia-Visso and Visso-Norcia mainshocks. I believe it is the same...

Yes, that is correct and we modified the caption accordingly.

- consider using a logarithmic y-scale for the seismicity rate - then the inset would not be needed (the exponential Omori decay would change to linear, which does not matter - it is used sometimes)

We would rather prefer to leave the plot as it is to better highlight the exponential Omori decay.

Supplements

Ln 82-83: I think 1-2 should read as 1-3 because DHwS is visible also in section 3. And what do you mean by SSE or NNW portions? I do not see DHwS in section 4.

Ln 84: I do not see DHwS in sections 6 and 7. Please mark DHwS by yellow polygon similar to Fig. 2.

Following this suggestion, to highlight DHwS we have labelled the DHwS with yellow polygons in the cross sections of Supplementary Fig. 3.

Supp. Fig.3:

- According to the map view, the section 3 is the same as the section shown in Fig. 2. But the seismicity is bit different, see e.g. the shape of the cluster C1. I think the reason is not the different width shown (2 or 1 km).

Yes, we agree with that. The reason of the differences is not only the width of the section (2 vs 1 km) but the selected time. In figure 2c of the main text we plotted the seismicity after Visso-Norcia for the entire catalogue until August 2017. In section 3 of Supplementary figure 3 we plotted the seismicity recorded from Amatrice mainshock to 40 days after Norcia mainshock. This is highlighted in the caption of figure 3.

- please explain the meaning of the red arrows in the caption

We have added a sentence in the caption specifying the meaning of the red arrows.

Supp Fig. 4: What is the difference of b) and c). Is c a zoom of b? Please explain in the caption

Agreed, we have explained the meaning of (c) in the caption.

Reviewer #2 (Remarks to the Author):

I am happy with the revisions that the authors have made and see no further changes needed for publication.

We thank the reviewer for the appreciation of our revision work carried out on the manuscript.

Reviewer #3 (Remarks to the Author):

I thank the authors for their great work on responding to each of my comments. The new supplementary materials are very valuable and help greatly the reader to understand the whole picture! I'm now convinced that this study deserves to be published in Nature Communications. I only have one or two comments that the authors might want to consider before the final proofs.

Congratulations again.

We thank again the reviewer for providing an in depth and constructive review of the first version of the manuscript. His/her comments significantly contributed to improve our work.

1) I think very interesting that the new figure/analysis provided by the authors in the rebuttal letter (answer to my comment 6: Interpretations) shows similar results i.e. that b -values in carbonates are lower than in TE, even situated away from the fault plane. It would imply that the on-fault behavior might not be so important compared to the rheology. Thus this is affecting somehow the results of the paper. The authors state that these results are out of the scope and need further investigations. I understand but since the authors already performed some measurements, I would add them in the paper (at least in the supplementary notes) and discuss a little bit further. I remind the authors that this is a transparent peer review process so reviewer's comments and answers will be available online. So to me it would worth it to describe it clearly in the paper/supplements before other people (mis)interpret the results. There is already a small dedicated part in the text and supplements so it won't take too much characters.

In the figure below, presented in the first revision round, b -values are evaluated for the different lithologies within the seismogenic volume. Indeed, the b -values in carbonates ($b=1.3$ for the area above C2-C3 and $b=1.43$ for the area below C2-C3) are lower than those estimated for the C1-C3 clusters in the TE (b in the range 1.66-1.81) and for DHwS (1.61-1.54). Our speculative interpretation of the lower value for carbonates is due to different environmental conditions at which the brittle/ductile transition may occur in comparison to the ductility of the anhydrites of the Triassic Evaporites. This is in part supported by the observation that the values within the carbonates observed below C2-C3 are still higher than on-fault (1.43 vs. 1.34 respectively) but slightly lower than in the TE, suggesting that brittle/ductile deformation in carbonates may occur at different environmental conditions than in the TE.

In conclusion, the significant differences in fault rheology and b -values, observed between on-fault seismicity and seismicity on TE, support our interpretation of distributed seismicity. Further studies on the rheology of carbonates at in situ conditions are required to further constrain our speculative arguments. Therefore, we prefer to focus our analysis only on distributed seismicity within TE. We have made this clear throughout the main text and the rebuttal, so that the reader will not misinterpret our results.

2) Very minor comments: please check the GB or US wording of "kilometre" and "hypocentre". I would use "kilometer" and "hypocenter".
 OK thanks, we have now consistently corrected our wording to English UK.